# Long-term whole blood DNA preservation by cost-efficient cryosilicification

Liang Zhou[1,7], Qi Lei[1,7], Jimin Guo[2,7], Yuanyuan Gao[1], Jianjun Shi[3], Hong Yu[1], Wenxiang Yin[1], Jiangfan Cao[1], Botao Xiao [1], Jacopo Andreo [4], Romy Ettlinger[5], C. Jeffrey Brinker [2], Stefan Wuttke [4,6] ✉ & Wei Zhu [1] ✉

Deoxyribonucleic acid (DNA) is the blueprint of life, and cost-effective methods for its long-term storage could have many potential benefits to society. Here we present the method of in situ cryosilicification of whole blood cells, which allows long-term preservation of DNA. Importantly, our straightforward approach is inexpensive, reliable, and yields cryosilicified samples that fulfill the essential criteria for safe, long-term DNA preservation, namely robustness against external stressors, such as radical oxygen species or ultraviolet radiation, and long-term stability in humid conditions at elevated temperatures. Our approach could enable the room temperature storage of genomic information in book-size format for more than one thousand years (thermally equivalent), costing only 0.5 $/person. Additionally, our demonstration of 3D-printed DNA banking artefacts, could potentially allow 'artificial fossilization'.

The deoxyribonucleic acid (DNA) represents a "genetic blueprint" of every individual containing important information about ancestry, health conditions, and traits[1–3]. Its successful safe and long-term preservation, i.e., DNA banking, provides several benefits to individuals, families, and society in general[4–8]: (i) tracing hereditary health conditions, assessing disease risks in families, monitoring potential symptoms, and treating diseases early[2,3,9]; (ii) serving as readily accessible forensic DNA evidence for personal identification and kinship analyses[4,10,11]; and (iii) its utilization as 'artificial fossilization' could address the challenge of continuously increasing numbers of endangered or extinct species[12,13].

Over 100 DNA banking companies and organizations worldwide offer DNA extraction from blood or saliva samples, and its storage for more than 50 years under cryoconditions, i.e., −80 °C to −164 °C, but at costs ranging from a hundred to a thousand dollars[4–7]. This limits the accessibility of DNA banking to non-wealthy consumers, and severely hinders its further expansion and development. Since DNA extraction prior to preservation, specialized equipment for snap-freezing, and inherently high costs of cryopreservation at ultra-low temperatures make DNA banking so expensive[5–7], in situ sample preservation with on-demand extraction and long-term room temperature storage represents a desirable goal[5,7,14–16]. For in situ preservation, formalin-fixation and paraffin-embedding became the most common methods[16–19]. However, formalin-fixation induces fragmentation and lesions of the DNA which reduce the yield of extracted DNA and lead to undesired sequence artifacts that interfere with DNA profiling[17–19]. For cost-efficient storage, so-called flinders technology associates (FTA) cards were developed containing reagents that promote cell lysis, protein denaturation, and facilitate the subsequent entrapment of released nucleic acids[19–21]. FTA cards provide a compact room temperature storage system[19], that protects DNA against different stresses including ultraviolet (UV) radiation, oxidation, and nucleases. But their cotton-based cellulose paper is degradable by cellulolytic enzymes and microorganisms or can naturally degrade, compromising safe DNA storage over decades[20–22].

[1]MOE International Joint Research Laboratory on Synthetic Biology and Medicines, School of Biology and Biological Engineering, South China University of Technology, Guangzhou 510006, P. R. China. [2]Center for Micro-Engineered Materials and the Department of Chemical and Biological Engineering, The University of New Mexico, Albuquerque, NM 87131, USA. [3]Science and Technology on Advanced Functional Composites Technology, Aerospace Research Institute of Materials & Processing Technology, Beijing 100076, P. R. China. [4]BCMaterials, Basque Center for Materials, UPV/EHU Science Park, 48940 Leioa, Spain. [5]School of Chemistry, University of St. Andrews, St. Andrews, United Kingdom. [6]Ikerbasque, Basque Foundation for Science, Bilbao, Spain. [7]These authors contributed equally: Liang Zhou, Qi Lei, Jimin Guo. ✉e-mail: stefan.wuttke@bcmaterials.net; weizhu86@scut.edu.cn

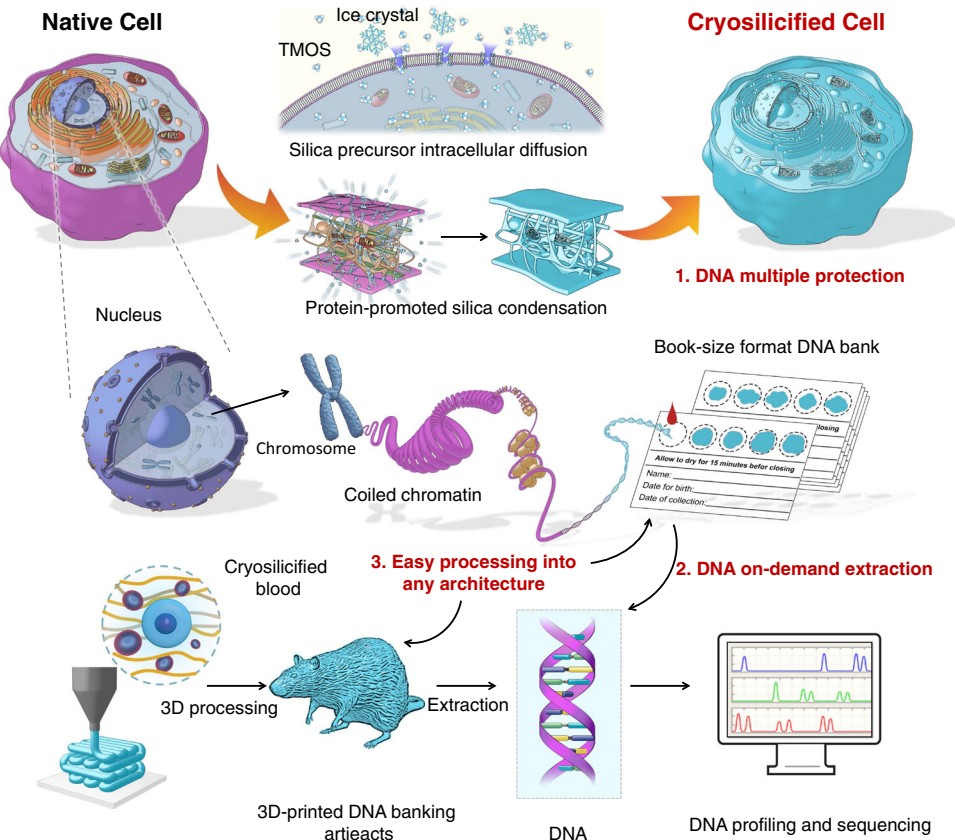

**Fig. 1 | Schematic representation of the cell cryosilicification process and the construction of whole blood DNA banking.** The silica precursor diffuses into the cell and nucleus under dormancy, and subsequently, condenses under the catalysis of cellular protein. This self-limited silica condensation on sub-cellular templates preserves the entire cellular architectures, mimicking the hermetic sealing of DNA in natural fossils. Through simple drop casting or three-dimensional (3D) printing technologies, whole-blood genomic DNA banking could be constructed for cryosilicified blood samples with various substrates for further on-demanded DNA profiling and sequencing.

Using mechanically strong and chemically inert inorganic silica that is stable for up to million years[23,24], emerged as an attractive strategy, namely biosilicification, to protect the activity of biomolecules from environmental stresses including heat, desiccation, lytic enzymatic exposure[23–25]. Purified DNA encapsulated in amorphous silica is resistant to chemical attack, high temperatures, and aggressive radical oxygen species (ROS)[26–30], but its extraction and purification are still time-consuming and expensive[4,5]. Consequently, researchers have focused on silicifying cells in situ[23,24]. However, encapsulating cells in siliceous exoskeletons is insufficient to prevent intracellular biomolecule autolysis, so pre-treatment remains essential[23,31,32].

To overcome the weaknesses of existing silicification processes, here we develop a simple "freezing cells in amorphous silica" strategy at −80 °C without fixation (Fig. 1). Using this method, DNA is immobilized inside the cell/silica composites, mimicking the hermetic sealing of naturally occurring fossils, and enabling the preservation of the entire cellular DNA and its native properties at room temperature, potentially for many centuries. We evaluate the resulting stability of the DNA against external stressors, i.e., ROS species or UV radiation, and accelerated aging. Additionally, we successfully construct whole blood genomic DNA banking with different substrates, i.e., filter paper-based cards or 3D-printed artifacts, fulfilling all essential requirements for future applications.

## Results
### Cryosilicification of whole blood cells
To cryosilicify cells, fresh anticoagulant peripheral blood was mixed with a pH 3 isotonic saline solution containing 15% w/w cryoprotectant hydroxyethyl starch and 50 mM silicic acid, i.e., Si(OH)₄ derived from the hydrolysis of tetramethyl orthosilicate (TMOS), at room temperature. The pH value of 3 was deliberately chosen, because then TMOS hydrolyzes rapidly and forms mainly uncharged Si(OH)₄ monomers, while their condensation to siloxane oligomers is suppressed[23,32]. After immersing cells into this silicification solution, the silicic acid species diffuse into the cell and successively exchange with water within hydrogen-bonded interfacial water networks surrounding biomolecular interfaces[23,32]. The mixture was kept at −80 °C for 24 h, thawed at room temperature, and rinsed with a phosphate-buffered saline solution (Fig. 2a). During the freezing step, the formation of nanosized ice crystals permeabilizes the cellular membranes and Si(OH)₄ further concentrates, accelerating its adsorption at extra- and intracellular biomolecular surfaces. Thereafter, the amino acids of proximal proteins catalyze the progressive, self-limiting condensation of silicic acid. This "freezes" the entire cell in nanoscopic amorphous silica coatings (Fig. 2b−e)[23,32].

To qualify and quantify the silica incorporation, fourier-transform infrared (FTIR) spectroscopy and inductively coupled plasma-optical emission (ICP-OES) spectroscopy were performed. After cryosilicification, additional bands were observed in the FTIR spectrum at 1080, 950, 790, and 450 cm⁻¹, which are attributed to asymmetric stretching of Si-O and Si-OH, symmetric stretching of Si-O, and bending vibration of Si-O-Si, respectively (Fig. 2f). The two bands at 1645 and 1525 cm⁻¹, which can be assigned to stretching modes of amide I and II of cellular proteins, remain unchanged during the cryosilicification. Also, both cryosilicified and native cell samples showed a similar SDS-PAGE pattern as shown in Supplementary Fig. 1,

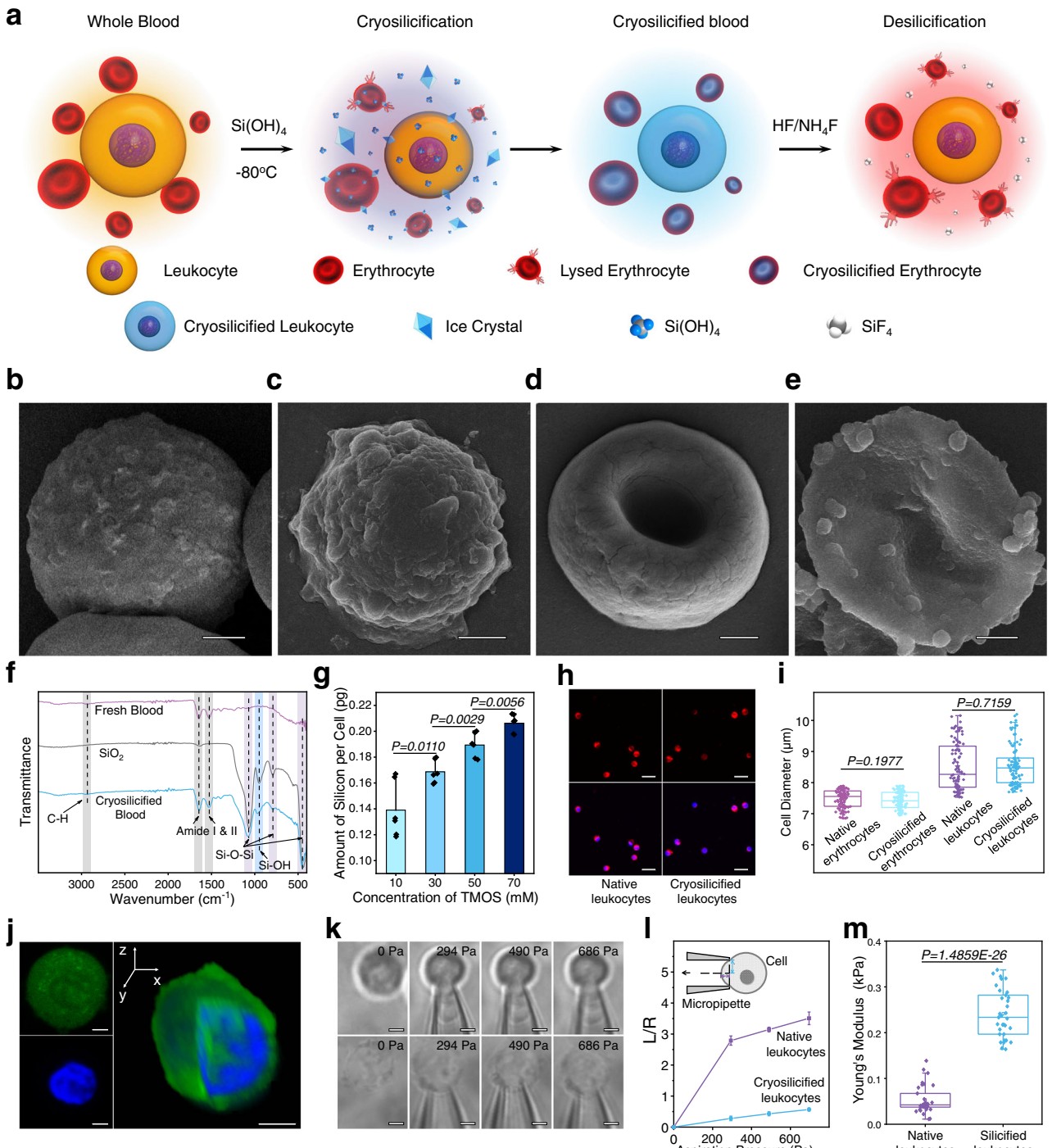

**Fig. 2 | In situ cryosilicification of whole-blood cells. a** Schematic representation of the procedure of cell cryosilicification and desilicification. **b–e** SEM images of **b** native leukocyte, **c** cryosilicified leukocyte, **d** native erythrocyte, and **e** cryosilicified erythrocyte, Scale bars = 1 μm. A representative image of three biological replicates is shown. **f** FTIR spectroscopy of native blood (purple), silica (gray), and cryosilicified blood (blue). A representative spectrum of three biological replicates is shown. **g** Silicon content of cryosilicified blood prepared with different silica precursors concentration determined by ICP-OES. Six biological replicates are shown. **h** Fluorescent microscopy images of native and cryosilicified leukocytes. Cell nucleus was stained by Hoechst 33342 (blue), and the cell membrane was stained by DiD (red), Scale bars = 20 μm. A representative image of three biological replicates is shown. **i** Cell diameter of leukocyte and erythrocyte after cryosilicification. Box and whiskers represent mean ± 25–75 percentile, center values 7.393,

7.242, 7.871, and 8.016, successively, $n = 102$ cells of three biological replicates. **j** 3D confocal microscopy images of cryosilicified leukocyte. Cell nucleus was stained by Hoechst 33342 (blue), and silica was stained by FITC-modified silane (green) during synthesis, Scale bars = 2 μm. A representative image of three biological replicates is shown. **k** Bright-field images of native and cryosilicified leukocyte under different aspiration pressures, Scale bars = 5 μm. A representative image of three biological replicates is shown. **l** Aspiration length changes of native and cryosilicified leukocyte under different aspiration pressures. Three biological replicates are shown. **m** Young's modules of native and cryosilicified leukocytes. Box and whiskers represent mean ± 25–75 percentile, center values 0.042 and 0.234, $n = 33$ biological replicates. Statistical significance was calculated with two-tailed Student's $t$ test. Data are presented as mean values ± SD. Source data are provided as a Source Data file.

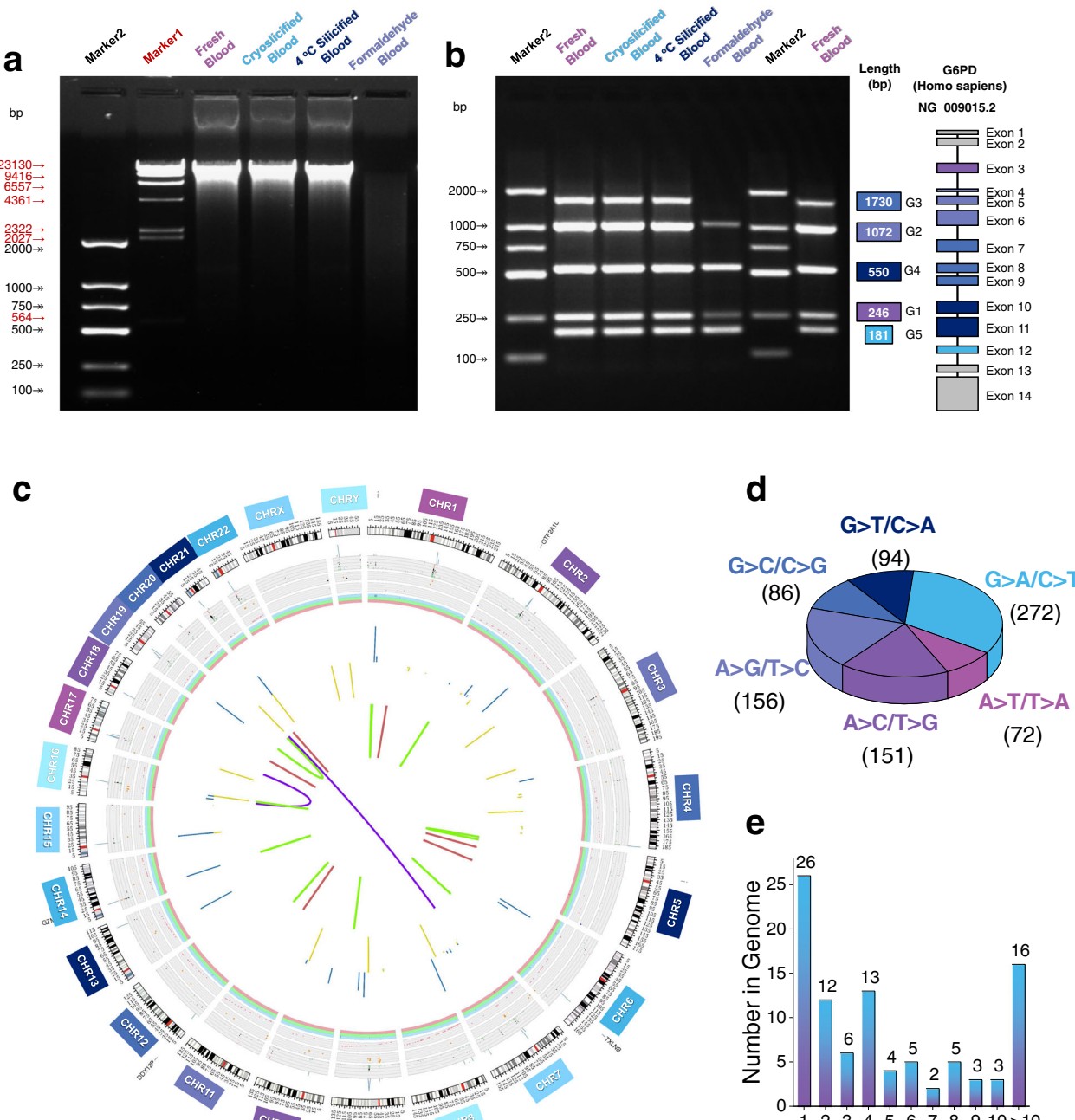

**Fig. 3 | Whole blood DNA preservation. a** Gel electrophoresis of DNA extracted from fresh blood, cryosilicified blood, 4 °C silicified blood, and formaldehyde-fixed blood samples. A representative image of three biological replicates is shown. **b** Gel electrophoresis of the amplified target genomic fragments (G1–G5) from fresh blood, cryosilicified blood, 4 °C silicified blood, and formaldehyde-fixed blood. A representative image of three biological replicates is shown. **c** Circos plot summary of mutations between fresh and cryosilicified blood samples in all human chromosomes (1–22+XY). **d** SNP mutations between fresh and cryosilicified blood samples. **e** InDel mutations between fresh and cryosilicified blood samples.

indicating the unchanged (primary/secondary) protein structures and proteins only serve as catalyst[32,33]. Quantification of silica content by ICP-OES spectroscopy elucidated that higher silica precursor concentrations result in a higher silica content of cryosilicified blood, and that a concentration of 50 mM results in ~2.55 mg silica per mL of blood (Fig. 2g).

Bright-field and fluorescent microscopy showed that cryosilicified cells exhibit uniform size distribution without cell aggregation, which is not the case for typical sol-gel encapsulation procedures (Fig. 2h). Counting 1 million cells revealed that only 66.7% of erythrocytes, but with 96.6% almost all leukocytes were preserved (Supplementary

Fig. 2). The large portion of erythrocytes, which were lysed or damaged, is irrelevant for DNA banking because they have no cell nucleus to store genetic information, as preserved leukocytes do[34–36]. In addition, flow cytometry yielded similar values for forward scatter and side scatter before and after cryosilicification (Supplementary Fig. 3), confirming that the dimensions and structure of individual blood cells were maintained (Fig. 2b–e, Fig. 2i).

As single, individual cells were successfully immobilized, the silica distribution within these cryosilicified cells could be assessed with energy-dispersive X-ray spectroscopy elemental mappings (Supplementary Fig. 4) and showed a homogeneous distribution of carbon,

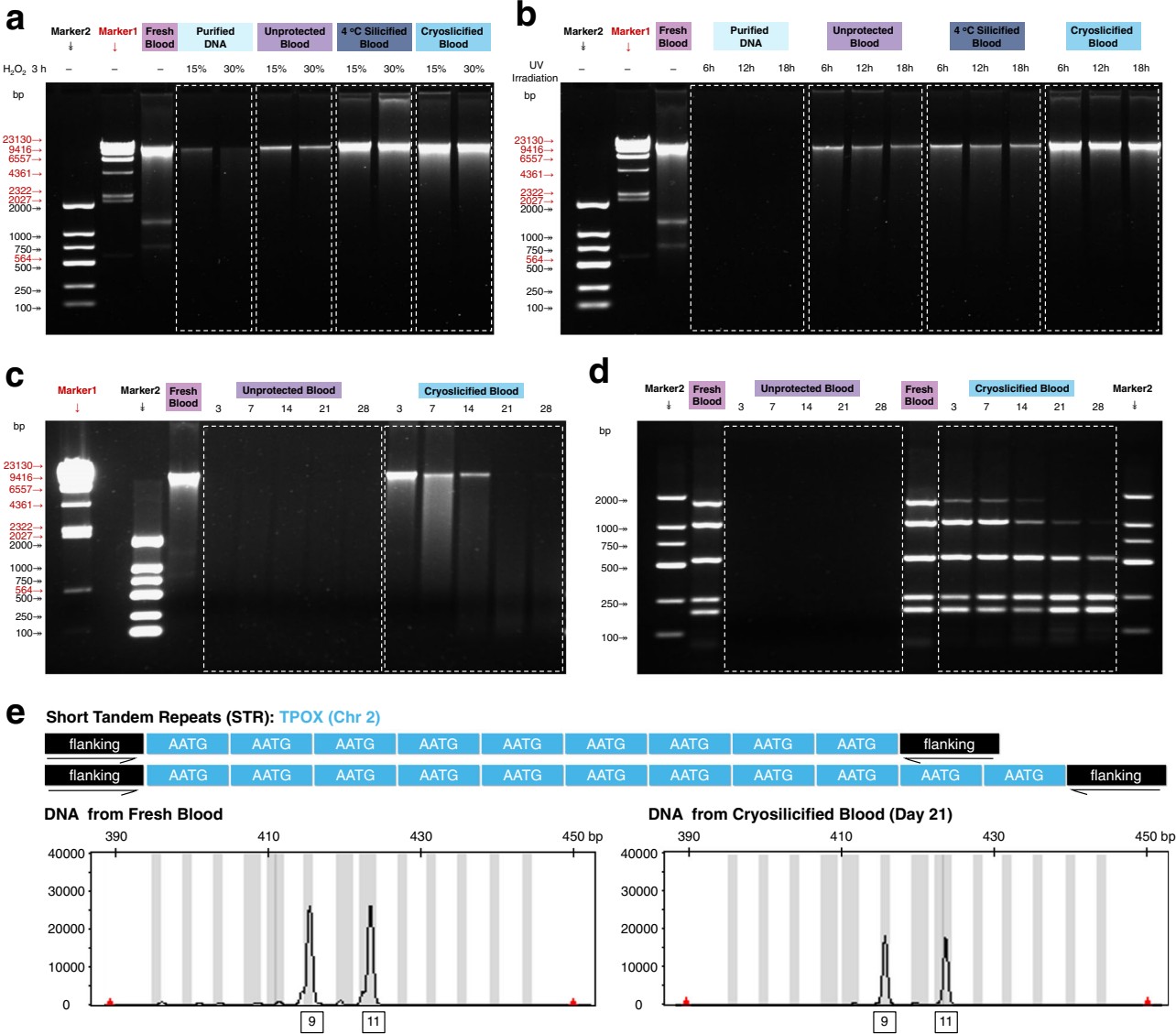

**Fig. 4 | Robust and long-term DNA preservation. a, b** Gel electrophoresis of DNA extracted from fresh blood sample and purified DNA, unprotected blood, 4 °C silicified blood, and cryosilicified blood samples after **a** $H_2O_2$ and **b** UV treatment. A representative image of three biological replicates is shown. **c** Gel electrophoresis of DNA extracted from a fresh blood sample, and unprotected blood and cryosilicified blood samples aging at 70 °C and 60% RH. A representative image of three biological replicates is shown. **d** Gel electrophoresis of the amplified target genomic fragments (G1–G5) from fresh blood sample, unprotected blood, and cryosilicified blood samples after aging at 70 °C and 60% RH. A representative image of three biological replicates is shown. **e** STR analysis of DNA from fresh blood samples and cryosilicified blood samples after aging 21 days at 70 °C and 60% RH.

oxygen, and silicon. To enhance visualization of the silica distribution, fluorescein isothiocyanate (FITC)-modified silane was utilized for cryosilicification, and then Hoechst 33342 was used to stain the nucleus. The confocal fluorescence microscopy images verified that silica diffused throughout cells forming siliceous cytoskeletons, which differs significantly from cells silicified at 4 °C, where siliceous exoskeletons were observed (Supplementary Fig. 5). Z-stack confocal microscopy images further revealed that the green fluorescence silica signal was even visible inside the nucleus, demonstrating the successful entry into the cell and nucleus (Fig. 2j, Supplementary Fig. 6). To verify the successful intracellular and intranuclear silicification, cryosilicified leukocytes were microtome-sliced and analyzed by TEM. No perturbation of cellular membrane or intracellular organelles was observable, and each organelle was coated with a uniform silica shell (Supplementary Fig. 7) at nanoscale (Supplementary Fig. 8). Consequently, our "freezing cells in amorphous silica" approach outperforms traditional encapsulations in exoskeletons because the entire

intracellular and intranuclear architecture was immobilized. Potentially preventing intracellular self-digestion of biomolecules by enzymes, this fulfills a prerequisite for long-term biomolecule preservation and DNA banking[23].

By determining Young's modulus of cryosilicified leukocytes using a micropipette aspiration technique (Supplementary Fig. 9)[37], another essential requirement for DNA banking was assessed: mechanical robustness against ambient stress and the ability to maintain architectural integrity over long periods of time[32]. When the pressure on the cell cortex of native leukocytes was increased from 0 to 294 Pa, they progressively deformed as they were extruded into the pipette until their surface adopted a hemisphere morphology inside the pipette (Fig. 2k, l). In contrast, cryosilicified leukocytes retained their original shape without significant deformation up to a pressure of 688 Pa. These stronger mechanical properties can be attributed to their siliceous cytoskeletons and are also reflected in Young's modulus: finite element analysis of the experimental configuration yielded

a value of 248 Pa for cryosilicified leukocytes, which is five times higher than that of 52.8 Pa for native leukocytes (Fig. 2m).

## Evaluation of DNA preservation

To prove that cryosilicification effectively preserves genetic information, the DNA was extracted from cryosilicified samples, systemically evaluated using a series of molecular assays, and compared to samples that were prepared by 4 °C silicification and formaldehyde-fixed. Per mL of blood samples, 6.26 µg and 5.93 µg DNA were extracted for cryosilicification and 4 °C silicification, corresponding to 76.7% and 72.6% of the yield of fresh blood samples, respectively (Supplementary Fig. 10). Nevertheless, formaldehyde-fixed samples yielded only 1.74 µg DNA, which is almost 3.6 times less compared with cryosilicified samples. Using gel electrophoresis, the DNA fragments were resolved: cryosilicification and 4 °C silicification samples showed the same bright and legible band at ~8 kbp as fresh blood samples, while formaldehyde-fixed samples only displayed a blurred and almost invisible band (Fig. 3a). Hence, unlike formaldehyde-fixed, cryosilicification proceeds without perturbating the DNA[17,18].

Additionally, target genomic fragments were amplified from extracted DNA using polymerase chain reaction (PCR). Based on the Homo sapiens chromosome X, i.e., G6PD (Supplementary Fig. 11) a DNA sequence containing 13 exons and encoding glucose-6-phosphate 1-dehydrogenase, five pairs of primers (Supplementary Table 1) were designed in different locations to obtain target DNA fragments with the lengths of 181, 246, 550, 1072, and 1730 bp[38]. According to their original sequence orders, they were marked from G1 to G5 (Fig. 3b). Gel electrophoresis revealed that all five DNA fragments were amplified for cryosilicification and 4 °C silicification samples. Their G3 (1730 bp) relative intensities (Supplementary Fig. 12) of 0.97 and 0.94, respectively, were again comparable to those of fresh blood samples. Interestingly, it was still possible to visualize four fragments from formaldehyde-fixed samples (Fig. 3b), although their extracted DNA degraded almost completely (Fig. 3a). However, their G3 relative intensities (0.13) were much lower than that of cryosilicified and fresh blood samples (Supplementary Fig. 12), emphasizing that formaldehyde-fixed causes DNA damage.

Encouraged to verify that cryosilicification could even preserve an entire personal genome, whole genomic sequencing (WGS) was performed on cryosilicified and fresh blood samples (Supplementary Table 6). The circos plot in Fig. 3c summarizes the mutations between cryosilicified and native blood in all human chromosomes (1–22+XY). With only 831 single nucleotide polymorphisms (SNPs) mutations (Fig. 3d) the error rate after cryosilicification corresponds to $2.68 \times 10^{-7}$, which is less than that of Taq DNA polymerase ($>10^{-6}$), and suggests that no perturbation took place for the entire genome[39,40]. Moreover, 95 insertion or deletion (InDel) mutations within a length range of 2–50 bp were observed (Fig. 3e). Since the blood sample was from the same person, these InDel errors could have been induced by multiple consecutive appearances of the same nucleotide[41]. Our evaluation of DNA preservation by in situ whole blood cryosilicification demonstrated that it safely 'freezes' the entire personal genome in silica.

## Stability studies

To prove safe, long-term DNA preservation, the resistance of cryosilicified cells to environmental genotoxic chemicals (ROS) and radiations (UV) that adversely affect the genome stability was assessed and compared to purified DNA and unprotected blood samples[25,30]. After the exposure to 15% $H_2O_2$ for 3 h, cryosilicified samples still showed a bright genomic DNA at ~8 kbp, while the band of purified DNA and unprotected blood samples was dim (Fig. 4a). The DNA extraction yielded 6.16 µg per mL of cryosilicified sample (Supplementary Fig. 13), which corresponds to 98.4% of the yield before exposure. Even when cryosilicified samples were exposed to non-diluted 30% $H_2O_2$ solution

for 3 h, identical results were obtained and 5.95 µg DNA could be extracted (Supplementary Fig. 13). PCR amplification successfully yielded all target genomic fragments (G1–G5) from the G6PD genomic DNA sequence (Supplementary Fig. 14). Moreover, only SNPs mutations were observed in $H_2O_2$-treated cryosilicified blood samples (error rate of $1.87 \times 10^{-7}$) (Supplementary Fig. 15, Supplementary Table 7). This impressive ROS resistance most likely results from the diffusion barrier caused by the dense siliceous cytoskeleton[23,42,43].

Since UV radiation usually also damages DNA during its long-term storage[25], cryosilicified samples were irradiated with 254 nm UV light (0.4–0.5 mW/cm²). Comparing the relative intensities of the band at ~8 kbp after 6 h to fresh blood samples, gave <0.04, 0.32, 0.37, and 0.75 for purified DNA, unprotected blood, 4 °C silicification and cryosilicified samples, respectively (Fig. 4b). The resistance for cryosilicified cells was attributed to the high UV-absorption of the cellular components of leukocytes and erythrocytes in the range of 200–300 nm (Supplementary Fig. 16). Even after 18 h of UV exposure, the relative intensity of cryosilicified blood samples remained as high as 0.65, while that of unprotected blood and 4 °C silicification samples were lower than 0.26 (Fig. 4b). The amount of extracted DNA per mL of cryosilicified blood dropped slightly from 6.26 µg before the UV treatment to 6.20, 5.86, and 5.15 µg after 6, 12 and 18 h, respectively (Supplementary Fig. 17). From WGS analysis, even with long-time UV exposure (18 h), only 617 SNPs mutations were found in UV-treated cryosilicified blood sample (error rate of $2.16 \times 10^{-7}$) (Supplementary Fig. 18, Supplementary Table 7), indicating negligible perturbation of the entire genome. Similar to diatoms, the dense siliceous cytoskeleton might thereby act as optical filter, protecting the encapsulated DNA from UV and excessive wavelength intensities[42,43].

Being resistant to external stressors, the stability of cryosilicified cells was assessed with accelerated aging tests at 70 °C and 60% relative humidity (RH). While all other samples lost their characteristic DNA band within 3 days, the cryosilicified samples remained stable and showed a visible band even after 14 days (Fig. 4c, Supplementary Fig. 19). As a control no significant degradation of DNA was also observed from cryosilicified blood samples during the storage at room temperature for 68 days (Supplementary Fig. 20). Evidently, their siliceous cytoskeleton effectively delays the degradation of the fragile DNA. The employed silicic acid-concentration directly impacts the yield of extracted DNA: after 7 days of aging, blood cryosilicified with 10, 30, 50, and 70 mM silicic acid yielded 1.64, 1.94, 2.94, and 3.08 µg DNA per mL, respectively (Supplementary Fig. 21). Note that, for comparison, the preservation of whole blood DNA on filter cards (regular filter card and Whatman® FTA® card) was also investigated. As shown in Supplementary Fig. 22 and Supplementary Fig. 23, for the fresh prepared samples, 5.85 µg and 6.18 µg DNA were extracted from the blood onto regular filter card and Whatman® FTA® card, respectively, which were comparable to the cryosilicified blood samples (6.26 µg) in Supplementary Fig. 10. The integrity of the extracted DNA was also confirmed by the bright band at ~8 kbp shown in the gel electrophoresis, which was similar to that of fresh blood samples and cryosilicified blood samples. However, after 14 days of aging at 70 °C and 60% RH, for DNA samples from the blood onto regular filter card and Whatman® FTA® card, the genomic DNA band (8 kbp) almost disappeared and the target genomic fragments (>246 bp) could not be amplified by PCR (Supplementary Fig. 24, Supplementary Fig. 25). In contrast, the cryosilicified blood samples clearly outperformed all the other samples: even after 14 days of aging, the genomic DNA band was preserved, indicating our cryosilicification technology was beneficial for safe whole-blood DNA preservation.

To verify genetic information preservation, the target genomic fragments (G1–G5) of the G6PD sequence were amplified by PCR. Gel electrophoresis confirmed that purified DNA and unprotected blood samples were completely degraded after 3 days (Fig. 4d and Supplementary Fig. 26). For formaldehyde-fixed and 4 °C silicification

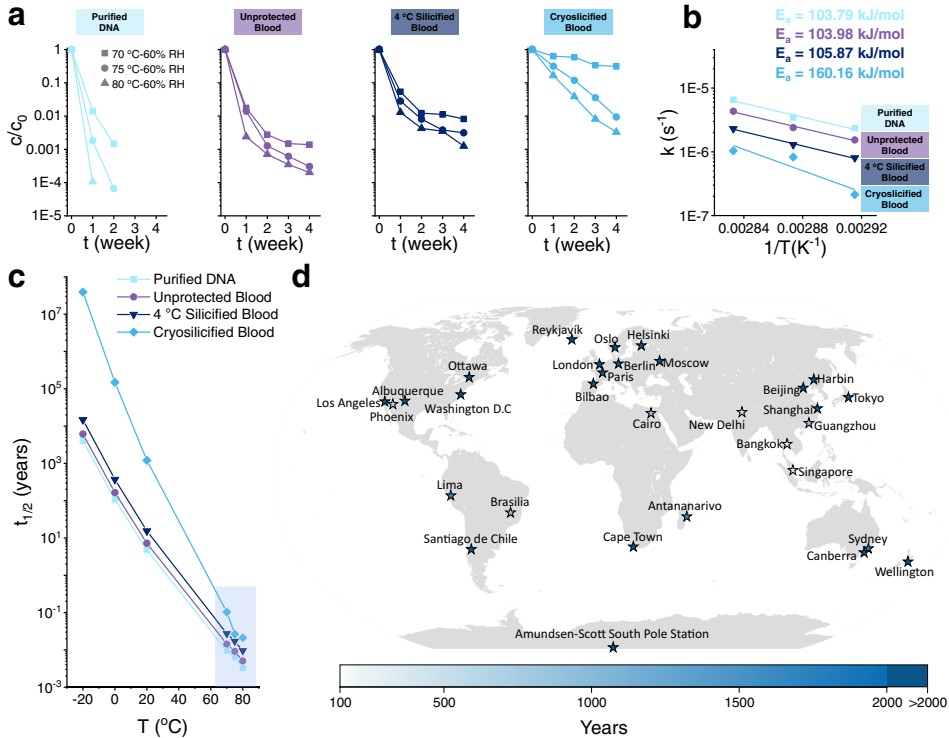

**Fig. 5 | Degradation kinetics of DNA storage. a** Concentration of target genomic fragments (G6) from purified DNA, unprotected blood, 4 °C silicified blood, and cryosilicified blood samples after aging at 70, 75, 80 °C and 60% RH. **b** Activation energies of target genomic fragments (G6) from purified DNA, unprotected blood, 4 °C silicified blood, and cryosilicified blood samples after aging at 70, 75, 80 °C, and 60% RH. **c** Half-lives extrapolated from the experimental decay rate constants. **d** Estimated half-lives of DNA from cryosilicified blood samples based on the average annual temperature of cities.

samples, the G2 and G3 band vanished and the G5, G1, and G4 band became blurred after 7 and 14 days, respectively (Supplementary Fig. 27 and Supplementary Fig. 28). The disappearance of the G2 and G3 bands can be attributed to the preferential degradation of large DNA fragments. The cryosilicified blood samples clearly outperformed the other samples: even after 14 days of aging, all five DNA fragments were amplified and showed their bands (Fig. 4d). Moreover, from WGS analysis, very low SNPs mutations with 670, 698, and 724 were also found in cryosilicified blood samples after 7, 14, and 28 days of aging, respectively, demonstrating the robust, long-term preservation of personal genetic information (Supplementary Table 8, Supplementary Figs. 29–31).

Moreover, the mainstay of forensic DNA analysis, i.e., short tandem repeat (STR) typing, was used to investigate the stability of cryosilicified cells[10,11]. Although complete STR profiles were obtained for unprotected blood samples after 21 and 28 days of aging, their average peak height significantly decreased in all loci, especially in larger-sized amplicons, indicating serious DNA degradation (Supplementary Figs. 32–34). In contrast, the STR profile of cryosilicified cells showed 100% concordance with fresh blood samples (Fig. 4e).

To quantify the degradation rate, the target DNA fragment G6 of each sample was analyzed by quantitative polymerase chain reaction (qPCR) after 7 days of aging. While 63% of G6 was detected for cryosilicified samples, only 1.41%, 1.75%, and 5.3% remained intact for purified DNA, unprotected blood, and 4 °C silicification samples, respectively (Fig. 5a). Even after 28 days of aging, 31% of the G6 remained of cryosilicified samples (Fig. 5a), underlining the effectiveness of cryosilicification. Based on our results and assuming first-order kinetics of degradation[27–29], the decay rate constants at 70 °C were calculated for purified DNA, unprotected blood, 4 °C silicification, and cryosilicified blood samples as $2.34 \times 10^{-6}$, $1.54 \times 10^{-6}$, $8.03 \times 10^{-7}$, and $2.13 \times 10^{-7}$ s$^{-1}$, respectively (Fig. 5b, Supplementary Table 4). Featuring only 9 and 14% of the degradation rate of purified DNA and

unprotected blood samples, respectively, cryosilicification offer a better protection against DNA degradation, which could arise from the stabilization of DNA-binding proteins[44]. The protein-catalyzed silica condensation stabilized the DNA-protein complex and formed a dense siliceous cytoskeleton to immobilize the cell structure[23,32].

To extrapolate the accelerated aging data to realistic storage temperatures, Arrhenius-type activation energies ($E_A$) were calculated based on the temperature dependence of the decay rates for purified DNA, unprotected blood, formaldehyde-fixed, 4 °C silicification, and cryosilicified blood samples (Degradation kinetics of DNA storage, Supplementary Table 5):[27–29] 103.79, 103.98, 105.87, and 160.16 kJ/mol, respectively. The extrapolated storage half-life for cryosilicified samples at 20 °C was about 1208 years, increasing the stability 250- and 167-fold compared to purified DNA and unprotected blood samples (Fig. 5c). At lower temperatures even longer storage times are possible: 3867, 53663, and 39 million years at 14.9 °C (the average temperature of the earth), 4 °C and −20 °C, respectively. Predicting real-world storage based on the average annual temperature of cities, yielded more than 1000 years for temperature lower than 20.8 °C (Fig. 5d and Supplementary Table 9).

## Development of DNA banking

To enable compact room temperature storage, we developed a paper-based DNA banking protocol (Fig. 6a). Following our cryosilicification method with 50 mM of silicic acid, the cryosilicified cells were dropped onto the filter paper-based blood cards after thawing and dried at room temperature. This protocol is neither time-consuming nor expensive, as it does not require DNA extraction, specialized equipment or ultra-low cryopreservation temperature. The protocol itself takes about 1 day for completion with a total hands-on time of <20 min, and after long-term storage, approximately one hour is needed for DNA extraction and purification. Such blood cards make it possible to preserve the genetic information of 1000 persons per 100 pages in

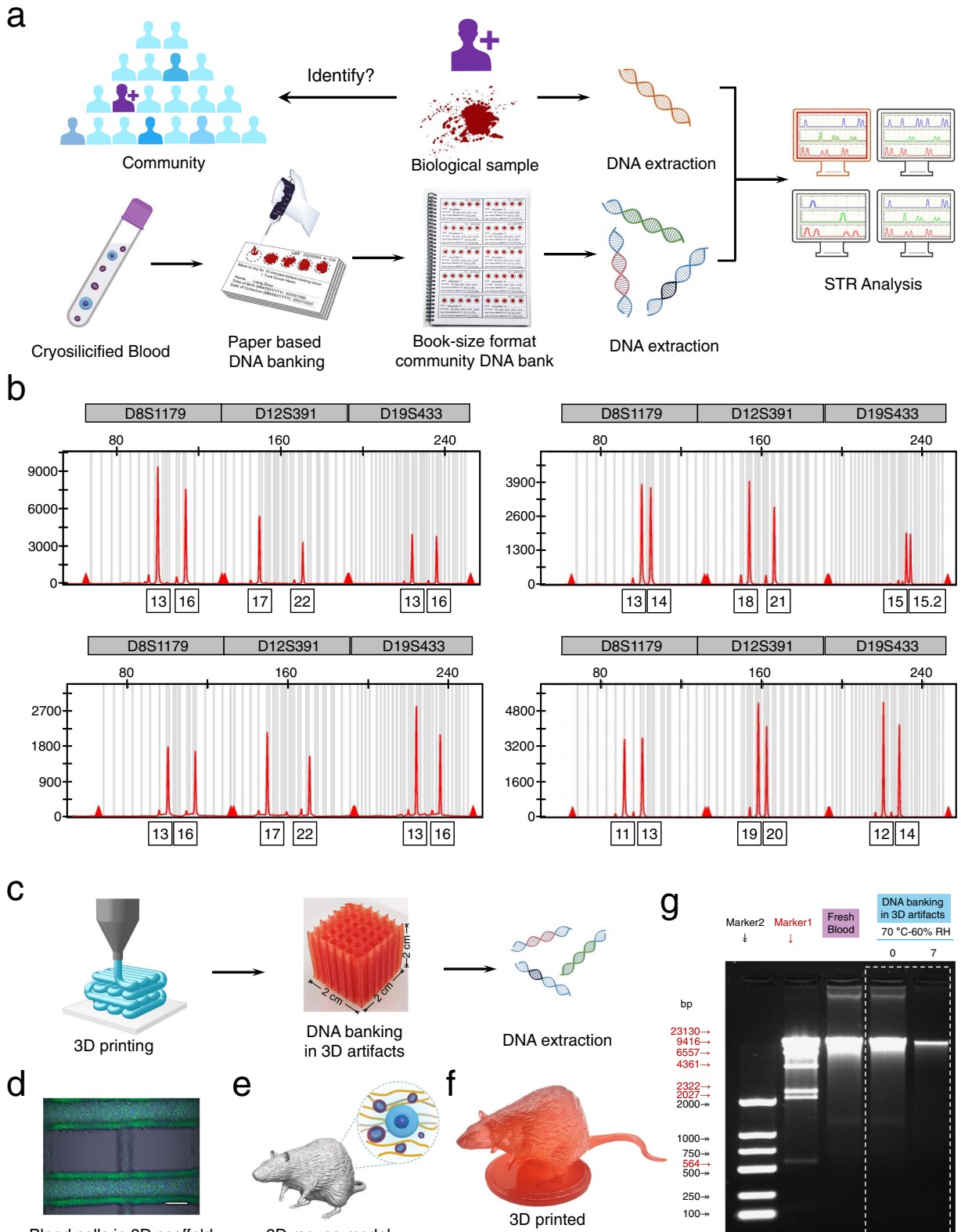

**Fig. 6 | Construction of whole blood DNA banking with paper-based cards or 3D-printed artifacts. a** Schematic representation of the preparation of paper-based whole blood genomic DNA banking and its application for forensic identification. **b** STR analysis of DNA extracted from fresh blood samples and paper-based whole blood DNA bank after aging for 28 days at 70 °C and 60% RH. **c** Schematic representation of the fabrication of whole blood genomic DNA banking with 3D-printed artifacts (middle, scale bars = 2 cm) and on-demanded DNA extraction. **d** Microscopy images of cryosilicified leukocytes and erythrocytes in 3D-printed whole blood DNA banking scaffold; Cell nucleus was stained by Hoechst 33342 (blue), and cell membrane was stained by DiO (green), Scale bars = 200 μm. **e** 3D-printed mouse model. **f** 3D-printed mouse figurine with embedded cryosilicified mouse blood samples. **g** Gel electrophoresis of DNA extracted from fresh blood, cryosilicified blood, and 3D-printed DNA banking artifacts before and after its aging for 7 days at 70 °C and 60% RH. A representative image of three biological replicates is shown.

book-size format (Fig. 6a and Supplementary Fig. 35). Importantly, for one person, the chemical costs of our procedure were about 0.2 $ and a total cost was less than 0.5 $, a reduction of 1000 to 10,000 times compared to conventional DNA banking.

Our blood cards represent a promising method for creating a forensic DNA database, which could be very helpful to identify missing persons and to develop investigative leads to assist law enforcement[4,10,11]. To demonstrate this, a small DNA database was generated by preparing the blood cards of three volunteers (denoted as volunteer I, II, and III). Their long-term storage was mimicked by accelerated aging at 70 °C and 60% RH for 28 days. The forensic identification was simulated by extracting DNA from one unknown fresh blood sample of one of the three volunteers and the three blood cards. Similar to real-world forensic analysis all extracted DNA samples were analyzed by STR typing (Supplementary Fig. 36–38), as this is compatible with degraded DNA, which is often found in evidentiary biological samples[11]. High-quality STR results were obtained for all blood cards, which identified that the sample of volunteer II is 100% in concordance with the fresh blood sample (Fig. 6b and Supplementary Fig. 39, Supplementary Table 10. Consequently, even after 28 days of accelerated aging, which equals ~1000 years of natural aging, it was undoubtedly clear that volunteer II provided the unknown fresh blood sample. STR analysis of blood cards, which were prepared with unprotected blood samples, revealed a complete DNA degradation after the simulated long-term storage (Supplementary Fig. 40), underlining the relevance of cryosilicification for the construction of forensic DNA databases.

As storing family genetic information or developing 'artificial fossilization' might require a higher degree of design freedom in the substrates used to symbolize traits, we also employed 3D printing technologies to create very complex artifacts for DNA banking (Fig. 6c and Supplementary Fig. 41)[26,45]. The cryosilicified samples were mixed with transparent 3D printer ink, then shaped using 3D printing and embedded in simple 3D porous DNA banking scaffolds (Fig. 6d and Supplementary Fig. 42). For DNA extraction, the 3D printer ink was simply dissolved, and the released cryosilicified cells were collected: from a 3D DNA banking scaffold 6.1 μg DNA were recovered (Supplementary Fig. 43), corresponding to 96.9% of the embedded cryosilicified samples. The integrity of the extracted DNA was confirmed by the bright band at ~8 kbp shown in the gel electrophores, which was similar to that of fresh blood samples (Fig. 6g). Even after 7 days of accelerated aging, 2.8 μg DNA could be extracted from the 3D DNA banking scaffold (5 g) (Supplementary Fig. 43). To illustrate the creative possibilities, we designed further 3D-printed DNA banking artifacts and a mouse figure storing its own 'genetic blueprint' (Fig. 6e, f and Supplementary Fig. 44). This approach combines the genetic information and its 3D appearance, coupled with the enhancement/ augmentation of the stability of DNA in blood cells to pass the harsh conditions of 3D printing, including high pressure in 3D printing, UV light in resin curing, etc, which is beneficial for identifying or describing species in multiple dimensions, and could be important in fighting species extinctions[45].

## Discussion

In summary, we demonstrated that bio-cryosilicification process is a facile, powerful technique to preserve the precious information of whole blood cells for the lifetime of numerous generations. This approach is based on cheap chemicals and a simple procedure, but it facilitates highly effective 'freezing' of DNA within thermally stable amorphous silica, avoiding ultra-low temperature storage and associated 'cold chain' problems. The stability, WGS analysis, and accelerated aging studies revealed that cryosilicified cells are extremely robust against external stressors, i.e., ROS, UV radiation, and long-term exposure to humid conditions at elevated temperatures. This highlights the potential of cryosilicification for the safe, long-term

storage of genomic information. Additionally, utilizing various substrates for DNA banking, i.e., filter paper-based cards or 3D-printed artifacts, expands application scenarios, which could satisfy societies' ever-growing need to preserve DNA compactly and cost-efficient. Looking forward, 'artificial fossilization' by cryosilicifying the genomic information of endangered or already extinct species could pave the way to halting species extinctions or perhaps even reversing them in the future.

## Methods

### Ethical statement

The study protocol was approved by the ethics committee, Guangzhou Women and Children's Medical Center, Guangzhou, China (Protocol 201928300). All procedures regarding human subjects have been reviewed and approved by the Ministry of Science and Technology of the People's Republic of China. All the participants understand and accept in detail the inspections and information to be provided. All participants' written consents were obtained. All participants met the following inclusion criteria: (1) age 18–40 years, Han Chinese; (2) at least 50 kg; (3) generally healthy by self-report. The exclusion criteria were as follows: (1) pregnancy or lactation; (2) cold and flu symptoms the day of collection; (3) infections within two weeks prior to collection; (4) symptoms of a heart condition within the six months prior to collection. Human blood was acquired from healthy donors with their informed consent. All blood samples were collected by experienced phlebotomists and stored in KWS vacutainer blood collection tubes (KWSMedical, Shijiazhuang, China) containing 1.5 mg of EDTA per mL of blood for anticoagulation purposes. The purification of whole blood was carried out using Ficoll density gradient centrifugation procedure.

### Animal

All animal procedures followed the criterion of the Institutional Animal Care and Use Committee (IACUC) of the Animal Experiment Center of South China University of Technology (Guangzhou, China) and were conducted following institutional approval (Protocol 2021010). All animal experimental procedures were implemented according to the Regulations for the Administration of Affairs Concerning Experimental Animals approved by the State Council of People's Republic of China, and the Guidelines for the Care and Use of Laboratory Animals by the Ministry of Science and Technology of the People's Republic of China. Housing conditions included 12-h dark/light cycle (light 6 a.m. to 6 p.m.), ambient temperature 24 °C, and humidity 50%.

### Reagents

All chemicals and reagents were used as received. TMOS, ethylenediaminetetraacetic acid (EDTA), hydroxyethyl starch (HES) and sodium chloride (NaCl) were purchased from Aladdin Bio-Chem Technology (Shanghai, China). FITC, (3-aminopropyl)-triethoxysilane (APTES) were purchased from Macklin Biochemical (Shanghai, China). Ethanol, methanol, formaldehyde solution (36.5–38% in $H_2O$), hydrogen peroxide ($H_2O_2$, 30%), 10× phosphate-buffered saline (10× PBS, pH 7.4), 1× Tris-acetate-EDTA buffer (1× TAE), Agarose G-10, GelRed® nucleic acid gel stain, DNA ladder (100-2000 bp, Marker 2), paraformaldehyde (PFA, 4%) and Ezup Column Blood Genomic DNA purification kit were purchased from Sangon Biotech (Shanghai, China). Taq PCR Master Mix (2×), Hoechst 33342, DiD, DiO, BeyoFast™ SYBR Green qPCR Mix (2×) were purchased from Beyotime Biotechnology (Shanghai, China). Ficoll-Paque™ PLUS, buffered oxide etch (BOE, 6:1) were purchased from Sigma-Aldrich (St. Louis, MO, USA). Lambda DNA/Hind III maker was purchased from Thermo Fisher Scientific (Waltham, MA, USA, Marker 1). SPI-Pon 812-Araldite 6005 epoxy embedding kit was purchased from SPI Supplies (West Chester, PA, USA). PowerSeq® 21 System was purchased from Promega (Madison, WI, USA). SunP Gel S1 was purchased from SunP Biotechnology (Beijing, China). UV-6017 was purchased from Ao Wei Digital Science and Technology Co., Ltd

(zhuhai, China). Milli-Q water with a resistivity of 18.2 MΩ was obtained from Aquaplore 2 S water purification system.

## Characterization

Scanning electron microscopy (SEM) analyses and energy-dispersive X-ray spectroscopy (EDS) elemental mappings were performed on a Zeiss Merlin field-emission scanning electron microscope at an accelerating voltage of 20.0 kV. Transmission electron microscopy (TEM) imaging were carried out using a Thermo Scientific™ Talos L 120 C TEM at 120 kV. FTIR spectra were measured using Thermo Nicolet Nexus 670 FTIR Spectrometer at the resolution of 8 cm$^{-1}$ in the wavenumber region 400–4000 cm$^{-1}$. Confocal fluorescent images were obtained on the Leica TCS SP8 confocal microscope using the Leica LAS X software. Flow cytometry measurements were obtained on the BD Accuri C6 plus flow cytometer. Silica content was determined using a Thermo Scientific™ iCAP 7200 ICP-OES. The DNA concentration was measured using the Thermo Scientific™ NanoDrop 2000 spectrophotometer. Gel electrophoresis images were captured using the gel imager BLT GV 5000 P system and the band relative intensities of these were analyzed using the BioAnaly software. Ultraviolet–visible (UV-Vis) absorption spectra were measured using Sunpbiotech UV-2600 spectrophotometer. Shenzhen Kings 3D Printing ALPHA-CPM2 and JS-450 were used to Whole blood genomic DNA banking in 3D-printed artifacts. We confirm that there is no third-party content including figures, stock photos, and clip arts in our manuscript.

## Silicification solution preparation

The silicification solution was prepared following the previous study with slight modification[46]. Briefly, 100 μL of 1 M HCl was added to 100 mL of 0.9% saline (154 mM NaCl) solution to prepare pH 3 saline solution. Then, 15 g HES was dissolved in 100 mL pH 3 saline solution. After full dissolution, 750 μL TMOS were added to achieve 50 mM silicification solution.

## Whole blood cell cryosilicification

200 μL whole blood was added to a tube containing 1800 μL silicification solution. Then the tube was closed and inverted 40 times to ensure proper mixing of blood and silicification solution. Then the mixture was transferred to −80 °C for 24 h to allow the silicification process to take place. The silicified whole blood was thawed at room temperature, rinsed with 1× PBS for three times, and then stored in 1× PBS. To compare cell silicification techniques, cells were also silicified at 4 °C in silicification solution for 24 h.

## Fourier-transform infrared spectroscopy

The samples were grinded into powder and placed onto the small crystal area of the sample plate. The transmittance measurements were carried on Thermo Nicolet Nexus 670 FTIR Spectrometer at the resolution of 8 cm$^{-1}$ in the wavenumber region 400–4000 cm$^{-1}$.

## Inductively coupled plasma-optical emission spectrometer

The silicified whole blood samples were first digested with nitric acid. After digestion, the samples were diluted with water to a final volume of 2.0 mL. And then, ICP-OES were performed to quantify the silica incorporation.

## Flow cytometry analysis

Blood samples were rinsed with 1× PBS and then diluted with 1× PBS to a final concentration of $5 \times 10^6$ cells/mL. Flow cytometry measurements were obtained on the BD Accuri C6 plus flow cytometer.

## SEM imaging and EDS elemental mappings

The morphology and elemental analysis of blood cell samples were characterized using SEM. To prepare untreated blood cells for SEM imaging, the fresh blood cells were fixed in 4% paraformaldehyde (in 1× PBS) at room temperature overnight, rinsed with 1× PBS three times, and stored in 1× PBS before further dehydration. To prepare silicified blood cells for SEM imaging, the fresh blood cells were silicified, rinsed with 1× PBS two times, and stored in 1× PBS before further dehydration. Blood cell samples were dehydrated by sequential rinsing in deionized water, 30% ethanol, 50% ethanol, 70% ethanol, 90% ethanol, 100% ethanol for 10 min in each solution. SEM samples were prepared by drop casting. Briefly, dehydrated blood cell samples were suspended in 100% ethanol, and then dropped onto 5 × 5 mm substrates (glass slides or aluminum foil). The substrates were then mounted on SEM stubs using conductive adhesive tape (12 mm OD PELCO Tabs). Samples were sputter coated with a 10 nm layer of gold using an EMS 150 T Sputtering System. SEM images and EDS elemental mappings were acquired at 20 kV using MERLIN field-emission scanning electron microscope.

## FITC-APTES synthesis

In all, 20 mg of FITC was first dissolved in 4 mL of anhydrous methanol and then 20 μL of APTES was added. The mixture was stirred at room temperature in the dark for 24 h. The resultant FITC-APTES solution was stored at 4 °C for future use.

## Confocal microscopy imaging

The leukocytes were first stained with Hoechst 33342 and then silicified at 24 h at −80 °C or +4 °C using FITC-APTES mixed silicification solution (the molar ratio of FITC-APTES:TMOS is 1:200). The resulted FITC labeled silicified blood cells were rinsed with 1× PBS for three times and stored in 1× PBS. The silicified blood cell sample solution was added to the confocal dish for further imaging. Confocal images were acquired with a ×100/1.4NA oil objective in sequential scanning mode using the Leica TCS SP8 confocal microscope.

## Microtome and TEM imaging

The silicified blood cell samples were rinsed with deionized water for two times and then dehydrated in 30%, 50%, 70%, 90%, 100% ethanol (two times). Subsequently, the samples were further dehydrated in 1:1 ethanol/acetone, 1:3 ethanol/acetone, and 100% acetone (two times). After dehydration, the samples were infiltrate with SPI-Pon 812 resin/acetone (50/50) for 1 h, SPI-Pon 812 resin/acetone (75/25) for 12 h, and SPI-Pon 812 resin for 2 h (two times). The infiltrated samples were then embedded in SPI-Pon 812 resin at 60 °C for 48 h. Ultrathin sections of 80 nm were cut by an EM UC7/FC7 ultramicrotome and collected on copper TEM mesh grids covered with a holey carbon support film. The ultrathin sections were observed using a Talos L 120 C TEM with 120 kV accelerating voltage.

## Mechanical characterization

The Young's modulus was measured using micropipette aspiration following a previous study with slight modification[47,48]. Micropipettes were made by drawing capillary tubes with a pipette puller and were then fractured on a microforge to an inner diameter of approximately 5 μm. Then the micropipettes were coated with 1% agar to prevent cell adhesion. The cell samples were diluted using 1× PBS to a final density of 100,000 cells/mL. A total of 33 native leukocytes and 33 cryosilicified leukocytes were analyzed. During the analysis, cell samples were each aspirated by a micropipette and driven by a computer-controlled piezoelectric translator to go through an approach-contact-withdrawal cycle. The repeated manipulations were performed on a light microscope and monitored with a camera. The outline of the segment of the cell drawn into the micropipette was measured and calibrated with a video caliper system (resolution of 0.2 μm) in synchronization with the recorded time and applied pressure. Linear correlation analysis was performed between the calculated moduli and the cell diameters, as well as between the moduli and the ratios of the cell to micropipette diameter.

## Desilicification

The silicified blood cells (equivalent to cells in the 200 μL whole blood) were immersed in 1500 μL of BOE, or so-called buffered HF solution, for 10 min to etch the silica. [Caution! HF is highly toxic. Extreme care should be taken when handling HF solution and only small quantities should be prepared.] The desilicified blood cells were rinsed with 1× PBS for two times and stored in 1× PBS.

## Protein extraction

In order to avoid the interference of hemoglobin, erythrocyte ghosts were prepared by incubating the erythrocytes or cryosilicified erythrocytes in 0.25× PBS overnight and collecting by centrifugation ($12,000 \times g$, 10 min), and then the membrane proteins were extracted with the Membrane Protein Extraction Kit (Sangon). The whole proteins of leukocytes and cryosilicified leukocytes were extracted from the samples using the Tissue or Cell Total Protein Extraction Kit (Sangon). All the obtained proteins were dissolved in 5× Protein Loading Dye (Sangon) and then run on 8% SDS-PAGE gel in 1×Tris-Glycine running buffer using Mini-PROTEAN tetra system (BIO-RAD). The samples were run at 120 V for 1.5 h. The polyacrylamide gel was stained in Coomassie Brilliant Blue R250 Protein Stain Reagent for observation.

## DNA extraction

DNA from the blood cell samples was extracted using the Ezup Column Blood Genomic DNA Purification Kit following the standard protocol provided by Sangon Biotech. The DNA concentration was measured using the Thermo Scientific™ NanoDrop 2000 spectrophotometer.

## Gel electrophoresis

The integrity of the DNA was determined by gel electrophoresis[49]. Briefly, samples were loaded to 1% Agarose G-10 gel containing GelRed® nucleic acid gel stain. Then the gel was run on the Bio-Rad electrophoresis system at room temperature (90 V, 30 min) and visualized under UV light. The relative grayscale or relative intensity of the gel electrophoresis band was analyzed using BioAnaly software. The relative gray scale is expressed as relative intensity equal to the gel electrophoresis band grayscale of sample/the gel electrophoresis band grayscale of fresh blood.

## PCR analysis

PCR was performed to investigate the protection of DNA by amplification of chromosomally encoded housekeeping gene G6PD (Figure S9). Primer sequences are listed in Supplementary Table 1. Each PCR reaction was performed with a Bio-Rad C1000 Touch™ PCR system, using the following protocol: 25 μL of Taq PCR Master Mix (2×), 4 μL of primer mix, 6 μL of templates, and 15 μL of Milli-Q water. Cycling parameters for PCR are shown in Supplementary Table 2.

**Whole-genomic sequencing.** After performing DNA quality inspection and library construction, the WGS was carried on the genomic DNA extracted from cryosilicified and fresh blood samples. The human genome hg19 assembly version (from UCSC database) was taken as the reference sequence using the comparison software BWA[50]. Comparing the filtered Clean Reads to the reference genome, the BAM format comparison result file was generated. Then SAMtools software was used to sorts BAM files[51] and sequences that only aligned to the unique position of the genome were screen out using Picard (http://broadinstitute.github.io/picard/) markers. In addition, GATK was used to local realignment of sequences around the InDel and reduce SNV detection false positive[51]. After the above series of processing, the BAM documents of high accuracy comparison results for mutation detection are obtained.

## Tolerance toward ROS

The ROS tolerance test was performed on the blood cell samples toward $H_2O_2$. Briefly, blood cell samples (equivalent to cells in the 200 μL whole blood) were rinsed with 1× PBS, and then suspended in 1500 μL of 15% or 30% $H_2O_2$ at room temperature. After 3 h incubation, blood cell samples were rinsed with 1× PBS for two times, and then stored in 1× PBS for further DNA integrity analysis.

## Tolerance toward UV exposure

The tolerance toward UV exposure was performed following the previous study with slight modifications[52]. Briefly, blood cell samples (equivalent to cells in the 200 μL whole blood) were added to the UV transparent 12-well plate. The blood cell samples were placed in UV8AC11HJ UV chamber (CBIO, China) equipped with compact UV Lamps (254 nm, 0.4–0.5 mW/cm²). After different exposure time, DNA from blood cell samples was extracted and DNA integrity was analyzed.

## UV-Vis absorption spectrum

Blood cell samples (equivalent to 200 μL whole blood) or purified DNA sample (extracted from 200 μL blood) were rinsed with 1× PBS, and then diluted 500 times with 1× PBS. UV-Vis absorption spectra were measured using UV-2600 spectrophotometer at a wavelength of 200-700 nm.

## Accelerated ageing test

The accelerated ageing test was performed following the previous study with slight modifications[53–55]. Briefly, blood cell samples (equivalent to cells in the 200 μL whole blood) were transferred to the Eppendorf tubes and stored with an open lid in the CTHL-150 Temp&Humidity Chamber (STIK, China) at 60% RH and three different temperatures (70 °C, 75 °C, and 80 °C). After different storage time, the DNA from the blood cell samples was extracted for further analysis.

**Degradation kinetics of DNA storage.** The decay kinetics of DNA can be expressed by the equation:[53–55]

$$-\frac{dC}{dt} = kC \tag{1}$$

where C is the DNA concentration and k is the decay rate constant, which integrated gives:

$$lnC = -kt + lnC_0 \tag{2}$$

To quantify the degradation rate, the qPCR was used to analysis the DNA concentration of the target DNA fragment G6 (213 bp) from the G6PD genomic DNA sequence during the accelerated aging study. The logarithm of the DNA concentrations measured by qPCR was fitted versus the time (Fig. 6a), and then linear regression analysis was applied to determine the decay rates according to the above equation (Supplementary Table 4).

The Arrhenius equation can be expressed by the equation:[53–55]

$$k_T = \mathbf{A}e^{\frac{-E_A}{RT}} \tag{3}$$

where $E_A$ is the activation energy and $k_T$ is the degradation kinetic constant for a given temperature $T$, which also expressed as:

$$lnk_T = -\frac{E_A}{RT} + lnk_0 \tag{4}$$

The activation energy was obtained by fitting the kinetic data following the Arrhenius equation (Supplementary Table 5).

The degradation kinetic constant $k_T$ for a given temperature $T$ was calculated with the Arrhenius' equation:[53–55]

$$k_T = \mathbf{A}e^{\frac{-E_A}{RT}} \tag{5}$$

The half-life ($t_{1/2}$) at a certain temperature $T$ was calculated with the equation:

$$t_{1/2} = \frac{ln2}{k_T} \tag{6}$$

### Short tandem repeat analysis

STR analysis were performed using the PowerSeq® 21 System, a PCR kit targeting the gender locus Amelogenin and 20 autosomal STR loci, following the standard protocol provided by Promega. The extracted DNA from blood cell samples served as a template for the STR amplification. STR loci information is listed in Supplementary Table 3.

### QPCR analysis

qPCR was utilized for quantification of DNA concentrations at different time points during accelerated aging. Every data point was collected from an individual sample tube. To assess DNA degradation, samples were analyzed with a Bio-Rad CFX Connect Real-Time PCR system using a standard protocol: 10 μL of BeyoFast™ SYBR Green qPCR Mix (2×), 0.8 μL of 10 μM primer mix (G6, Forward primer 5′–3′: CTAAC-CACACACCTGTTCCCTC and Reverse primer 5′–3′: AGCCCACGAT-GAAGGTGTTTT), and 9.2 μL of templates. The cycling parameters for qPCR program were as follows: predenaturation at 95 °C for 3 min followed by 50 cycles of denaturing at 95 °C for 10 s, annealing at 61 °C for 20 s, and elongation at 72 °C for 30 s, and final elongation at 72 °C for 10 min. Relative sample quantification was accomplished by inter-polation from a standard curve, generated from DNA samples of known concentration. Reported values are averages from two technical replicates. Melting curve analysis was performed in the range of 65 °C to 95 °C with 0.5 °C per 5 s increment.

### Whole blood DNA banking on paper

The anticoagulant whole blood was collected and silicified with a 50 mM silicic acid silicification solution at −80 °C for 24 h. After thawing at room temperature, the cryosilicified blood samples were dropped directly onto the filter paper-based blood cards and dried at room temperature for DNA banking.

### Whole blood DNA banking in 3D-printed artifacts

Thermoresponsive 3D printing: The ALPHA-CPM 23D printer (Sunp-biotech, China) was used to fabricate 3D artifacts with whole blood DNA banking. Briefly, 200 μL cryosilicified blood cell samples were first suspended in 5 mL SunP Gel S1 (liquid at 4 °C but solid at 20 °C), which has been pre-cooled at 4 °C for 5 min. This solution was then loaded into a sterilized 1 mL syringe and set in the printer for printing in air. The 3D artifact was fabricated by forced extrusion (0.4 mm³/s) in a sterile atmosphere of 25 °C at 1.2 mm/s printing speed in a layer-by-layer fashion. After 3D bioprinting, these 3D artifacts were stored at room temperature for DNA banking. The main component of SunP Gel S1 is a temperature-responsive material, which begins to dissolve below 20 °C. For the 3D-printed artifacts using SunP Gel S1 (containing 200 μL of cryosilicified whole blood), 3D-printed artifacts were placed at 4 °C for 10 min-dissolution and washed with PBS (5000 rpm, 3 min) to remove the SunP Gel S1 and obtain the cryosilicified cells. Then the obtained cells were rinsed in 1.5 mL of BOE for 1 min to remove silica, and further washed with PBS three times before DNA extraction using the commercial kit. Light-induced 3D printing: 2000 μL cryosilicified blood cell samples were first suspended in 50 mL UV-6017. This mix-ture solution was loaded into JS-450 3D printer. After 3D bioprinting, these 3D artifacts were cured under UV light for DNA banking.

### Statistics and reproducibility

Data were presented as mean ± standard deviation. The data were analyzed by two-tailed unpaired $t$ test to calculate $p$ values for com-parisons between two groups using Graphpad Prism 8 and Origin 2018 software. Results were considered statistically significant when $p < 0.05$. No data were excluded from the analyses. The experiments were randomized. The Investigators were blinded to allocation during experiments and outcome assessment. At least three biological repli-cates were performed for all in vitro experiments unless otherwise indicated.

### Reporting summary

Further information on research design is available in the Nature Research Reporting Summary linked to this article.

## Data availability

The source data underlying Fig. 2f, g, i, l, m, 3a, b, d, e, 4a–c, d, 5a–c, 6g and Supplementary Figs. 1, 2, 8, 10, 12–31, 32–34, 36–38, 40 and 43 are provided as a Source Data file. The whole-genomic sequencing data are available at NCBI under Project PRJNA878673. Source data are pro-vided with this paper.

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

## Acknowledgements

This work was supported by the National Natural Science Foundation of China (21972047 to W.Z., 52003086 to Q.L.), Guangdong Provincial Pearl River Talents Program (2019QN01Y314 to Q.L.), the Program for Guangdong Introducing Innovative and Entrepreneurial Teams (2019ZT08Y318 to W.Z.), Natural Science Foundation of Guangdong Province, China (2021A1515010724 to Q.L.), China Postdoctoral Science Foundation (2020M672625, 2021T140213 to Q.L.), Science and Technology Project of Guangzhou, China (202102020352 to W.Z., 202102020259 to Q.L.), the Fundamental Research Funds for the Central Universities of China. The authors thank the support from the Guangzhou Women and Children's Medical Center and Laboratory

Animal Research Center of the South China University of Technology. S.W. acknowledges funding from the Basque Government Industry Department under the ELKARTEK and HAZITEK programs.

## Author contributions

L.Z., Q.L., J.G., J.B., S.W., and W.Z. conceived and designed the research. L.Z. performed DNA extraction, gel electrophoresis, PCR, and qPCR assays. Q.L. performed the CLSM and flow cytometry assays. Y.G., H.Y., and B.X measured Young's modulus of cells. W.Y., J.A., J.S., and J.C. designed and performed the 3D printing experiments. L.Z., Q.L., and J.G. analyzed and interpreted the data. Q.L., R.E., and J.G. organized and wrote the manuscript. J.A., J.B., S.W., and W.Z. revised the manuscript. All the authors discussed the results and commented on the manuscript.

## Competing interests

The authors declare no competing interests.
