## [Peer Review File · Nature Communications]

REVIEWER COMMENTS

Reviewer #1 (Remarks to the Author):

The manuscript by L. Zhou et al. describes a novel way of preserving DNA in whole blood samples. The technology builds on the preservation of DNA in silica nanoparticles (e.g. Paunescu et al. Nat. Protocols 2013), but in the present case the technology is not applied to purified DNA, but to the DNA still present in blood cells. This brings the advantage that the DNA does not have to be purified from the blood at the timepoint of collection, but rather at the timepoint of analysis. This has a cost advantage, if significantly more samples are collected than analyzed.

While, to my understanding the idea is new and innovative, several questions remain unanswered:

Three conceptual issues:

A) the idea of storing blood instead of DNA for later analysis is not new, and standardly applied for forensic samples. It is unclear where the advantage of the new method is compared to e.g. filter cards. A further drawback is the inability of the method to perturb organelles such as mitochondria, which also comprise valuable DNA

B) the storage of DNA in artefacts is certainly of interest. However, the advantage of the approach (not requiring DNA purification) seems small compared to the downsides using non-purified samples for the formation of artefacts (blood samples are regarded as highly infectious materials, purified DNA is not).

C) As the authors can only show the advantage of the method for the preservation of DNA, and no indication of the preservation of other components of the cells (proteins, RNA, metabolites), and as stated above there is already technology to store blood for later DNA analysis, the benefits of the new technology are somewhat unclear.

several issues in terms of scientific presentation and discussion:

- there is a collection of non proven and non-scientific statements in the text. This includes e.g. "amino acids of proximal proteins catalyse the progressive, self-limiting condensation of silicic acid." This is pure speculation. In the following sentence "in nanoscopic amorphous silica coatings (Fig. 2b-e)". The coatings in the images are several micrometers in scale, there is no proof of nanoscopic.

- line 111: "indicating that they do not perturb cellular protein structures". FTIR is not a suitable technology to assess the integrity of proteins structures!

- line 149: it is unclear why mechanical robustness against ambient stress is of any value when preserving DNA.

- instead of focusing on the WGS analysis and reporting error rates, the metric of yield is completely missing. How does the method compare in yield (DNA per ml of blood) vs. standard DNA extraction methods, and vs. the use of Whatman filter cards for blood/DNA storage?

- authors perform stability comparison not with the state of the art technologies, as e.g. storing blood on appropriate filters.

- line 232-260: all given quantification data in the text from samples aged at 70°C and 60% rel hum? Should be stated more clearly.

- the comparison around line 285 should be prepared in a more quantitative way, comparing three variants: a) blood cryosilification, then purification, b) blood on filter cards, then purification, c) first purification, then DNA encapsulation in silica, direct use after DNA release from silica.

- the study uses blood samples from volunteers. This usually requires ethical committee approval. This is not reported in the reporting summary.

- the authors present results for the STR typing of individual II, but not for the other samples. What is the reason for this?

- the conclusion has to be rewritten, as it is currently highly unscientific, and not a scientific discussion, but rather a sales argumentation.

Reviewer #2 (Remarks to the Author):

Zhou et al. demonstrate silicification of cells under cryogenic conditions for long-term preservation of genomic information. Their work tackles an important problem in storing biological samples without fixation or cold storage, performing flash freezing once then silicifying cells for storage. They demonstrate impressive retention of genomic information with minimal loss of sequence fidelity upon DNA extraction, compared with conventional fixation. They additionally impose silicified samples to harsh conditions including ROS, UV, and elevated temperatures. Finally, they 3D print their samples into various geometries, mimicking some authors in the field of DNA data storage, and also demonstrate sample storage using paper cards. Imaging and micropipette aspiration demonstrate silicification solidifies leukocytes, though it is somewhat unclear to what extent genomic DNA is actually protected in its entirety from harsh/damaging conditions.

While overall the principle of applying silicification to whole cells/leukocytes is attractive, compared with previous applications to extracted genomic DNA (by Grass et al.) or plasmids for DNA data storage (by Bathe et al.), my main reservation for this work in its current form is the limited demonstration of genomic sequence preservation following ROS/UV/aging, which are precisely the harsh conditions under which silicification is supposed to be useful. The current demonstrations of DNA extraction quantities and limited PCR amplification are helpful, but not illuminating as to the rigorous preservation of genomic DNA sequence, as the authors demonstrate using sequencing for their initial samples that are merely cryopreserved and silicified without harsh treatment/aging. In my view, detailed whole-genome sequence analysis should ideally be performed for their other samples that are treated harshly in order to validate the utility of their approach for this primary purpose of protection of genomic information.

Further, as a minor point, while it is “cute” that they 3D print their silicified cellular samples into 3D objects such as a rodent, this reviewer missed the point of why this is useful, first, and why this is innovative or technically challenging, second, and therefore worthy of incorporation into this scientific article.

Reviewer #3 (Remarks to the Author):

This work discusses the exploitation of bioinspired cell-silicification for stabilization of DNAs in the cell, which is a natural expansion of the applications of the reported silicification strategy. Although the work is not extremely innovative in the aspect of scientific development, its application proposed would deserve a publication in *Nat. Commun.* Some comments are as follows:

1. It is recommended to discuss the “cryosilicification” further, especially, in the aspect of mechanisms. Reference 23 (JACS, 2021) in the manuscript mentions the method at -80 oC (as ref. 114, Silicified Cell Replicas, Methods of Making, and Methods of Using. US Patent 20200276286), but it seems that the patent does not contain the cryosilicification method. Several issues also should be clarified: Why is the temperature of -80 oC selected? How do the silicic acid and its derivatives diffuse into the cell at the “frozen” condition? Are only organelles coated dominantly or are siliceous structures formed in the entire cytosol? Why are RBCs ruptured under the fabrication conditions (compared with their previous reports on RBC silicification)?

2. They mentioned the costly cryopreservation of extracted DNAs, but a control experiment with the extracted DNAs in silica matrices or silica-coated DNAs should be done. DNA extraction and subsequent silicification would not be highly costly.

3. It is not clearly discussed why and how the silica protects DNAs from degradation. In addition, the control experiment at room temperature should be done, like the previous report on proteins.

4. HF is used for desilicification. DNA stability under the desilicification process needs to be discussed along with justification of the HF use. Also, it should be stated how DNAs are extracted from paper-based and 3D printing-based samples: the manuscript just says “was simply dissolved”, and the supplementary materials do not contain the information.

Responses to reviewer's comments are as follows:

Reviewer #1:

The manuscript by L. Zhou et al. describes a novel way of preserving DNA in whole blood samples. The technology builds on the preservation of DNA in silica nanoparticles (e.g. Paunescu et al. Nat. Protocols 2013), but in the present case the technology is not applied to purified DNA, but to the DNA still present in blood cells. This brings the advantage that the DNA does not have to be purified from the blood at the timepoint of collection, but rather at the timepoint of analysis. This has a cost advantage, if significantly more samples are collected than analyzed. While, to my understanding the idea is new and innovative, several questions remain unanswered:

Response: We thank this reviewer for their careful consideration of our MS and for providing very useful criticisms, suggestions, and recommendations. Following the suggestion of the reviewer, we have carried out more substantiating experiments and have taken every effort to clarify the novelty of this work.

Three conceptual issues:

A) the idea of storing blood instead of DNA for later analysis is not new, and standardly applied for forensic samples. It is unclear where the advantage of the new method is compared to e.g. filter cards. A further drawback is the inability of the method to perturb organelles such as mitochondria, which also comprise valuable DNA

Response: This is an important comment. Following the suggestion, the preservation of whole blood DNA through standard protection schemes, including loading blood onto regular filter card and Whatman® FTA® card has been evaluated. For comparison, the preservation of whole blood DNA was also investigated on regular filter cards and Whatman® FTA® cards: from freshly prepared samples 5.85 and 6.18 µg DNA were extracted for regular filter cards and Whatman® FTA® cards (Fig. S22 and Fig. S23), respectively, which is comparable to the 6.26 µg of cryosilicified blood samples (Fig. S10). The integrity of the extracted DNA was also confirmed by the bright band at approximately 8 kbp shown in the gel electrophoresis, which was similar to that of fresh blood samples and cryosilicified blood samples. However, after 14 days of aging at 70°C and 60% RH, the genomic DNA band (8 kbp) of DNA samples from the blood of regular filter cards and Whatman® FTA® cards, almost disappeared and the target genomic fragments (>246 bp) could not be amplified by PCR (Fig. S24 and S25). In contrast, the cryosilicified blood samples clearly outperformed all the other samples: even after 14 days of aging, the genomic DNA band was preserved, indicating our cryosilicification technology was beneficial for safe whole blood DNA preservation. This information has been shown in the revised manuscript. (Please see page 13, paragraph 1, line 3-18)

Moreover, the organellar DNA (e.g. mitochondrial DNA) represents a mainstay of phylogenetics and evolutionary biology. Following the suggestion of the reviewer, we

detected the released DNA during cryosilicification process. As shown in **Table R1**, no DNA released during the cryosilicification process, indicating not only nuclear DNA but also organellar DNA could be preserved in cryosilicified cells. In current study, we mainly focused on the nuclear DNA, which encodes for the majority of the genome in eukaryotes. In our future work, we will investigate the preservation of organellar DNA.

Table R1. Release of leukocyte contents after cryosilicification

Name	Protein (mg/mL)	DNA (ng/ μ L)	RNA (ng/ μ L)
Supernatant	0.016 \pm 0.004	0.723 \pm 0.159	0.239 \pm 0.059
Leukocytes	0.710 \pm 0.019	30.367 \pm 1.626	18.267 \pm 0.306

*The supernatant was obtained by centrifugation at 3000 rpm for 5 min after cryosilicification

The protein was obtained by collecting the frozen cryosilicification Leukocytes and then sonicating (VCX130.SONICS. USA. Total time 5 min, ultrasonic 5 s, pause 5 s, ultrasonic power 80%).

DNA was extracted with Ezup Column Blood Genomic DNA Purification Kit by collecting the frozen cryosilicification Leukocytes

RNA was extracted with Total RNA Extractor (Trizol) kit by collecting the frozen cryosilicification Leukocytes

B) the storage of DNA in artefacts is certainly of interest. However, the advantage of the approach (not requiring DNA purification) seems small compared to the downsides using non-purified samples for the formation of artefacts (blood samples are regarded as highly infectious materials, purified DNA is not).

Response: Many thanks for this good question. The purified DNA samples contain only the genetic information based on their nucleotide sequence. The preservation of purified DNA samples as ‘artificial fossil’ might be constrained by the extent of existing knowledge. However, our cryosilicified blood samples contain more complete bioinformation (genomics and proteomics information), since there is no nucleic acids or proteins released during the cryosilicification process (**Table R1**). Although limited by current extraction technologies, some of these bioinformation might not be completely extracted, with the development of technologies, the more preserved bioinformation in cryosilicified blood samples will be disclosed. Therefore, we believe our ‘artificial fossil’ could be beneficial for identifying or describing species in multiple dimensions and important in fighting species extinctions.

Blood and body fluids have been considered as infectious materials due to the potential risk for infection from bloodborne pathogens (*Clin. Microbiol. Rev.* **2000**, *13*, 385-407). To address the safety concerns, the bioactivities of pathogens during cryosilicification process were investigated. As shown in **Figure R2**, **Figure R3**, **Table R2**, there are no bioactivities of both bacteria (*E. coli* K-12 MG1655) and virus (M13 Phage) after cryosilicification process. This indicated that even bloodborne pathogens might be carried in the blood samples, pathogens would lose their infectivity and virulence after cryosilicification process, demonstrating that cryosilicified blood samples are not potentially infectious with pathogens and therefore are not considered infectious materials.

Figure R2. Representative image of MG1655 (*E. coli*) bacteria colonies formed by MG1655 (*E. coli*) (A) without or (B) with cryosilicification and 7 days of storage at room temperature via the agar plate dilution method.

Figure R3. Representative image of XL1-blue (*E. coli*) bacteria colonies formed with M13 phage infection (kanamycin resistance gene). (a) Untreated phage infect XL1-blue, (b&c) Silicified phage (with 7 days storage at room temperature) infect XL1-blue, (d) Control XL1-blue without phage infection.

Table R2. Survival of different infection sources after cryosilicification

Name	Untreated	7 days at room temperature in an ultra-clean bench after cryosilicification
MG1655 (E. coli)		
Cfu/mL	$(2.4 \pm 0.5) * 10^9$	0
M13 phage		
Pfu/mL	$(1.0 \pm 0.1) * 10^{11}$	0

C) As the authors can only show the advantage of the method for the preservation of DNA, and no indication of the preservation of other components of the cells (proteins, RNA, metabolites), and as stated above there is already technology to store blood for later DNA analysis, the benefits of the new technology are somewhat unclear.

Response: Really good point. As shown in **Table R1**, there is no nucleic acids or proteins released during the cryosilicification process, indicating other components of the cells (proteins and RNA) might be preserved in cryosilicified cells. To future confirm the protein preservation, the SDS-PAGE was carried out to evaluate the cellular proteins before and after cryosilicification. The proteins in cryosilicified cells showed the similar SDS-PAGE pattern comparing those in native cells (**Figure S1**), indicating the preservation of native cellular proteins throughout the cryosilicification process. For the RNA and metabolites, we believe these could also be preserved in cryosilicified cells. However, due to the limitation of extraction technologies, we haven't got promising data to demonstrate the success of RNA and metabolites preservation. It is worth to note that we mainly focused on the whole blood DNA preservation in the current study. In our opinion, the preservation of other components of the cells is beyond the scope of the current study and will be the subject of future investigation. (Please see page 7, paragraph 1, line 2-5)

several issues in terms of scientific presentation and discussion:

- there is a collection of non proven and non-scientific statements in the text. This includes e.g. "amino acids of proximal proteins catalyse the progressive, self-limiting condensation of silicic acid." This is pure speculation. In the following sentence "in nanoscopic amorphous silica coatings (Fig. 2b-e)". The coatings in the images are several micrometers in scale, there is no proof of nanoscopic.

Response: The sentence of "amino acids of proximal proteins catalyse the progressive, self-limiting condensation of silicic acid." is based on previous publications (*ACS Nano* **2011**, *5*, 1401-1409; *Proc. Natl. Acad. Sci. U.S.A.* **2012**, *109*, 17336-17341; *J. Am. Chem. Soc.* **2021**, *143*, 6305–6322; *ACS Nano* **2022**, *16*, *2*, 2164–2175). In one of previous studies, arbitrary 3D protein hierarchical scaffolds have been created to investigate their ability to direct silica condensation (*ACS Nano* **2011**, *5*, 1401– 1409.). At pH 3, silica exists primarily as neutrally charged monosilicic acid and self-condensation to SiO₂ is limited. Under these conditions, the silicic acid monomers could diffuse throughout the scaffolds, interchange with interfacial water, and be condensed by the protein scaffolds that serve as silica condensation catalysts. Although it is generally thought that cationic species (e.g., polyamine or proteins with pI > 7) are necessary to promote silica condensation under physiological conditions, it is found that proteins with different identities, functional properties, and net charge (e.g., BSA, avidin, and lysozyme) all form protein/silica composites under pH 3 conditions (*ACS Nano* **2011**, *5*, 1401-1409; *J. Am. Chem. Soc.* **2021**, *143*, 6305–6322). Similar protein/silica composites formation were also observed in fixed and then room-temperature-silicified cell samples (*Proc. Natl. Acad. Sci. U.S.A.* **2012**, *109*, 17336-17341; *ACS Nano* **2022**, *16*, *2*, 2164–2175). Because catalytic proteins are occluded by silica deposition, silica bioreplication is naturally self-limiting, resulting in conformal

~10 nm thick silica deposits (*ACS Nano* **2011**, *5*, 1401–1409; *Proc. Natl. Acad. Sci. U.S.A.* **2012**, *109*, 17336-17341; *ACS Nano* **2022**, *16*, 2, 2164–2175). The references have been added in the revised manuscript.

To answer the questions from reviewer whether amorphous silica coatings is nanometers in scale, we further analyzed the transmission electron microscopy (TEM) cross-sectional images of cryosilicified cells and evaluated the thickness of silica coatings on the cell membrane. As shown in **Figure S7** and **Figure S8**, the thickness of the silica coatings on the plasma membrane and nuclear membrane is approximately 46 nm and 53 nm, respectively, demonstrating that the entire cell was ‘frozen’ in nanoscopic amorphous silica coatings. This information has been added in the revised manuscript. (Please see page 8, paragraph 1, line 10-12)

- line 111: "indicating that they do not perturb cellular protein structures". FTIR is not a suitable technology to assess the integrity of proteins structures!

Response: Agreed. The FTIR data show that the amide I and II bands, which depend mainly on primary and secondary protein structure, are unperturbed, but these bands are not as sensitive to tertiary and quaternary structure. It is worth to note that the complete sentence is “The two bands at 1645 and 1525 cm^{-1} , which can be assigned to stretching modes of amide I and II of cellular proteins, remain unchanged during the cryosilicification, indicating that they do not perturb cellular protein structures and only serve as catalyst.”. Here, we wanted to highlight that the cellular proteins only serve as catalyst but they are not involved in the silicification reactions, which is unrelated to the integrity of proteins. To avoid misunderstanding, we have weakened the protein structures statement in the revised manuscript. To investigate the integrity of cellular protein, the SDS-PAGE was added in the revised manuscript as shown in **Figure S1** to evaluate the cellular proteins before and after cryosilicification. The proteins in cryosilicified cells showed the similar SDS-PAGE pattern comparing those in native cells, indicating the preservation of native cellular protein throughout the cryosilicification process. This information has been shown in the revised manuscript. (Please see page 7, paragraph 1, line 2-5)

- line 149: it is unclear why mechanical robustness against ambient stress is of any value when preserving DNA.

Response: Many thanks for this good question. The mechanical robustness against ambient stress is important for the integrity of whole cell. If the cells were broken, the DNA might expose to environmental stresses and could be damaged by the microorganisms and nucleases. Moreover, the mechanical robustness is also benefit for the process of 3D printing, where the shear stress at the nozzle site would affect the integrity of whole cell (*Adv. Healthc. Mater.* **2016**, *5*, 326-333.). Avoiding shear stress-induced cell damage in 3D-bioprinting might help to improve the DNA preservation in 3D scaffold.

- instead of focusing on the WGS analysis and reporting error rates, the metric of yield is completely missing. How does the method compare in yield (DNA per ml of blood) vs.

standard DNA extraction methods, and vs. the use of Whatman filter cards for blood/DNA storage?

Response: Many thanks for this good question. Per mL of blood samples, 6.26 μg DNA were extracted for cryosilicified blood samples, corresponding to 76.7% of the yield of fresh blood samples (8.16 μg DNA), while blood samples storage on Whatman[®] FTA[®] card yielded 6.18 μg DNA (**Figure S10 and Figure S23**). This information has been shown in the revised manuscript. (Please see page 13, paragraph 1, line 3-18)

- authors perform stability comparison not with the state of the art technologies, as e.g. storing blood on appropriate filters.

Response: Many thanks for this comment. Following the suggestion, the preservation of whole blood DNA through different protection schemes, including loading blood onto regular filter card and Whatman[®] FTA[®] card has been evaluated. For comparison, the preservation of whole blood DNA was also investigated on regular filter cards and Whatman[®] FTA[®] cards: from freshly prepared samples 5.85 and 6.18 μg DNA were extracted for regular filter cards and Whatman[®] FTA[®] cards (**Fig. S22 and Fig. S23**), respectively, which is comparable to the 6.26 μg of cryosilicified blood samples (**Fig. S10**). The integrity of the extracted DNA was also confirmed by the bright band at approximately 8 kbp shown in the gel electrophoresis, which was similar to that of fresh blood samples and cryosilicified blood samples. However, after 14 days of aging at 70°C and 60% RH, the genomic DNA band (8 kbp) of DNA samples from the blood of regular filter cards and Whatman[®] FTA[®] cards, almost disappeared and the target genomic fragments (>246 bp) could not be amplified by PCR (**Fig. S24 and S25**). In contrast, the cryosilicified blood samples clearly outperformed all the other samples: even after 14 days of aging, the genomic DNA band was preserved, indicating our cryosilicification technology was beneficial for safe whole blood DNA preservation. This information has been shown in the revised manuscript. (Please see page 13, paragraph 1, line 3-18)

- line 232-260: all given quantification data in the text from samples aged at 70°C and 60% rel hum? Should be stated more clearly.

Response: Many thanks for this question. DNA degrades by the same single-strand break mechanism and the individual decay rates depend merely on the storage temperature and the water concentration within the vicinity of the DNA molecules (*Angew. Chem. Int. Ed.* **2015**, *54*, 2552-2555.). The accelerated aging condition (70°C and 60% RH) were chosen based on previous publications (*Angew. Chem. Int. Ed.* **2015**, *54*, 2552-2555; *Adv. Funct. Mater.* **2019**, *29*, 1901672). In this study, we found that 28 days of accelerated aging at 70°C and 60% RH is thermally equivalent to approximate 1000 years of natural aging. Therefore, unless otherwise specified, all aging condition in the text was 70°C and 60% RH. Notably, only in the degradation kinetics of DNA storage section (Section S23, Figure 5, Table S5), we aged blood samples at 75°C-60% RH and 80°C-60% RH. To avoid miscommunication, the information of aging condition has been specified in the revised manuscript.

- the comparison around line 285 should be prepared in a more quantitative way, comparing three variants: a) blood cryosilification, then purification, b) blood on filter cards, then purification, c) first purification, then DNA encapsulation in silica, direct use after DNA release from silica.

Response: Many thanks for this comment. Following the suggestion, the preservation of whole blood DNA through different protection schemes, including loading blood onto filter cards (regular filter card and Whatman® FTA® card) and encapsulating whole blood DNA in silica, has been evaluated. For comparison, the preservation of whole blood DNA was also investigated on regular filter cards and Whatman® FTA® cards: from freshly prepared samples 5.85 and 6.18 µg DNA were extracted for regular filter cards and Whatman® FTA® cards (**Fig. S22** and **Fig. S23**), respectively, which is comparable to the 6.26 µg of cryosilicified blood samples (**Fig. S10**). The integrity of the extracted DNA was also confirmed by the bright band at approximately 8 kbp shown in the gel electrophoresis, which was similar to that of fresh blood samples and cryosilicified blood samples. However, after 14 days of aging at 70°C and 60% RH, the genomic DNA band (8 kbp) of DNA samples from the blood of regular filter cards and Whatman® FTA® cards, almost disappeared and the target genomic fragments (>246 bp) could not be amplified by PCR (**Fig. S24** and **S25**). In contrast, the cryosilicified blood samples clearly outperformed all the other samples: even after 14 days of aging, the genomic DNA band was preserved, indicating our cryosilification technology was beneficial for safe whole blood DNA preservation. This information has been shown in the revised manuscript. (Please see page 13, paragraph 1, line 3-18)

For whole blood DNA encapsulation in silica, the samples were prepared following previous study with slight modification (*Nat. Protoc.* **2013**, *8*, 2440–2448). Instead of encapsulating small DNA fragments (100 bp) as previous study, to match our objective of whole blood DNA preservation, whole blood DNA (8 kbp) was encapsulated in silica (**Figure 3a**). Nevertheless, as shown in **Figure R1**, the genomic DNA from DNA encapsulation in silica vanished (blurred) even before storage. This unsuccessful DNA preservation might be due to our unskilled practice or the reported encapsulation strategy may not be suitable for the object of whole blood DNA. More studies in this part will be carried out in the future. This information has been shown in the revised manuscript.

- the study uses blood samples from volunteers. This usually requires ethical committee approval. This is not reported in the reporting summary.

Response: We apologize for this mistake. The missing Ethical Committee Approval has now been added in the reporting summary.

- the authors present results for the STR typing of individual II, but not for the other samples. What is the reason for this?

Response: Many thanks for this question. STR analysis of extracted DNA from the blood card after aging for 28 days at 70°C and 60% RH of three volunteers (denoted as volunteer I, II and III) was shown in **Figure S36, 37, 38**. The forensic identification was simulated by extracting DNA from one unknown fresh blood sample of one of the three volunteers and the three blood cards. Based on random choice, volunteer II was chosen to provide the unknown fresh blood sample. The experimental design of this simulated forensic identification was added as **Figure S39**.

- the conclusion has to be rewritten, as it is currently highly unscientific, and not a scientific discussion, but rather a sales argumentation.

Response: Many thanks for this comment. The conclusion has been rewritten in the revised manuscript.

Reviewer #2:

Zhou et al. demonstrate silicification of cells under cryogenic conditions for long-term preservation of genomic information. Their work tackles an important problem in storing biological samples without fixation or cold storage, performing flash freezing once then silicifying cells for storage. They demonstrate impressive retention of genomic information with minimal loss of sequence fidelity upon DNA extraction, compared with conventional fixation. They additionally impose silicified samples to harsh conditions including ROS, UV, and elevated temperatures. Finally, they 3D print their samples into various geometries, mimicking some authors in the field of DNA data storage, and also demonstrate sample storage using paper cards. Imaging and micropipette aspiration demonstrate silicification solidifies leukocytes, though it is somewhat unclear to what extent genomic DNA is actually protected in its entirety from harsh/damaging conditions.

Response: We thank this reviewer for their careful consideration of our MS and for providing very useful criticisms, suggestions, and recommendations. Following the suggestion of the reviewer, we have carried out more substantiating experiments and have taken every effort to clarify the novelty of this work.

While overall the principle of applying silicification to whole cells/leukocytes is attractive, compared with previous applications to extracted genomic DNA (by Grass et al.) or plasmids for DNA data storage (by Bathe et al.), my main reservation for this work in its current form is the limited demonstration of genomic sequence preservation following ROS/UV/aging, which are precisely the harsh conditions under which silicification is supposed to be useful. The current demonstrations of DNA extraction quantities and limited PCR amplification are helpful, but not illuminating as to the rigorous preservation of genomic DNA sequence, as the authors demonstrate using sequencing for their initial samples that are merely cryopreserved and silicified without harsh treatment/aging. In my view, detailed whole-genome sequence analysis should ideally be performed for their other samples that are treated harshly in order

to validate the utility of their approach for this primary purpose of protection of genomic information.

Response: Many thanks for this good suggestion. Following the suggestion, whole-genome sequence analysis has been performed on cryosilicified blood samples after H₂O₂ and UV treatment as well as after aging at 70 °C and 60% RH for different days. The summary of mutations between the cryosilicified blood with different treatments and the native blood in all human chromosomes (1-22+XY) was shown as a circos plot in **Figure 3c**. For insertion or deletion (InDel) mutations, 10, 25, 41, 100 InDel mutations within a length range of 10-50 bp were observed in cryosilicified blood samples and those after 7, 14, and 28 days of aging, respectively (**Figure S29c, 30c, 31c**), indicating the gradual DNA degradation during the aging process. Similar results were also observed in cryosilicified blood samples after H₂O₂ and UV treatments (**Figure S13 and Figure S17**). However, for single nucleotide polymorphisms (SNPs) mutations, compared with the native blood samples, 535 and 617 SNPs mutations were found in H₂O₂ and UV treated cryosilicified blood samples, respectively (**Figure S15 and Figure S18**). In addition, 670, 698, and 724 SNPs mutations were found in cryosilicified blood samples after 7, 14, and 28 days of aging, respectively (**Figure S29b, 30b, 31b**). It is noteworthy that after cryosilicification 831 SNPs mutations (error rate of 2.68×10^{-7}) were detected (Figure 3d), which is less than the error rate of Taq DNA polymerase ($>10^{-6}$). Similarly, the error rate of cryosilicified blood samples after harsh treatments were also low. This suggested no obvious perturbation took place for the entire genome during the cryosilicification process as well as preservation after harsh treatments, demonstrating that our *in situ* whole blood cryosilicification could preserve the entire personal genome in amorphous silica even after harsh treatments. This information has been shown in the revised manuscript. (Please see page 11, paragraph 2, line 17-18; page 12, paragraph 1, line 10-13; page 14, paragraph 1, line 6-9)

Further, as a minor point, while it is “cute” that they 3D print their silicified cellular samples into 3D objects such as a rodent, this reviewer missed the point of why this is useful, first, and why this is innovative or technically challenging, second, and therefore worthy of incorporation into this scientific article.

Response: We thank this reviewer for this critical comment. The DNA represents a ‘genetic blueprint’ and contains important information about ancestry, health conditions, and traits. For a family, the successful preservation of the DNA would allow to capture all this precious information which in turn could enable the tracing hereditary health conditions, the assessment of disease risks of a family, the monitoring of potential symptoms, and early treatment. Since the personal DNA bank, similar as coat of arms, is a source of information showing and tracing the membership of a family, the family genetic information storage requires more design freedom of substrates to symbolize traits and impress the viewer. It is known that the 3D printing technologies enables the easy creation of very complex geometries. Therefore, we constructed whole blood genomic DNA banking in 3D printed artifacts. Moreover, for endangered or extinct species, DNA banking could be utilized as an ‘artificial fossil’ to hold their bioinformation for the future studies. After several generations,

although the bioinformation of extinct species could still be preserved through cryosilicification technology, their morphology information may be lost. Therefore, creating realistic 3D printed replicas storing their own ‘genetic blueprint’ could be benefit for identifying or describing species in multiple dimensions. It also has vast possibilities for improving museum displays and sparking scientific curiosity museum visitors.

It is known that DNA in blood samples is fragile. High temperatures, strong pH swings, environmental genotoxic chemicals (ROS) and radiations (UV) all cause it to fragment, degrading the genetic information it encodes. The main technically challenge of 3D DNA banking scaffolds is how to enhancing/augmenting the stability of DNA in blood cells to pass the harsh conditions of 3D printing, including high pressure in 3D printing, UV light in resin curing, etc. Herein, our cryosilicification technology fulfills the essential criteria enabling the establishment of 3D-printed DNA banking artefacts.

Reviewer #3 (Remarks to the Author):

This work discusses the exploitation of bioinspired cell-silicification for stabilization of DNAs in the cell, which is a natural expansion of the applications of the reported silicification strategy. Although the work is not extremely innovative in the aspect of scientific development, its application proposed would deserve a publication in *Nat. Commun.*

Response: We thank this reviewer for their careful consideration of our MS and for providing very useful criticisms, suggestions, and recommendations. Following the suggestion of the reviewer, we have carried out more substantiating experiments and have taken every effort to clarify the novelty of this work.

Some comments are as follows:

1. It is recommended to discuss the “cryosilicification” further, especially, in the aspect of mechanisms. Reference 23 (*JACS*, 2021) in the manuscript mentions the method at -80 oC (as ref. 114, *Silicified Cell Replicas, Methods of Making, and Methods of Using*. US Patent 20200276286), but it seems that the patent does not contain the cryosilicification method. Several issues also should be clarified: Why is the temperature of -80 °C selected? How do the silicic acid and its derivatives diffuse into the cell at the “frozen” condition? Are only organelles coated dominantly or are siliceous structures formed in the entire cytosol? Why are RBCs ruptured under the fabrication conditions (compared with their previous reports on RBC silicification)?

Response: First of all, we apologize for the mistake in *JACS* 2021 paper. The citation in *JACS* paper will be updated by the following publication (*Nat. Biomed. Eng.* **2022**, *6*, 19-31.).

Your questions about cryosilicification are all brilliant questions, although they are beyond the scope of the current study, long-term whole blood DNA preservation. The

mechanism of cryosilicification is still not fully understood and we are currently focusing on it. Based on our preliminary experiments, we conjecture that the cryosilicification could happen under any temperature below the freezing point of water (0°C). The diffusion of silicic acid and its derivatives into the cell does not happen at the “frozen” condition but in the “thaw” process. During the formation of ice crystals, the microphase separation might happen and the silicic acids could concentrate closed to the cell surfaces. Then the small ice crystals, due to ice-recrystallization during the thaw process, damage to cell membrane shortly, which helps silicic acid transports across the cellular membrane. Once within the cytosol, silica precursors could be captured and concentrated by crowded protein environment via hydrogen bonding and other noncovalent interactions. It worth to note that, in this text, the cryosilicified blood samples was thawed at room temperature instead of 37°C, which provides enough time for the silica precursors to cross the cellular membrane, diffuse throughout the entire cytosol, and be condensed by the protein scaffolds. Due to the very low silica amount of cryosilicified cells, it is hard to achieve the silica cell replicas without structural collapse as previous study by simple calcination (*Proc. Natl. Acad. Sci. U.S.A.* **2012**, 109, 17336-17341; *J. Am. Chem. Soc.* **2021**, 143, 6305–6322). Although right now we do not have direct evidences to support the silicification of entire cytosol, the osmotic resistance from previous study (*Nat. Biomed. Eng.* **2022**, 6, 19-31.) and biomolecules preservation experiments (**Figure S5**) indicated the success of whole cell cryosilicification.

For the RBC silicification, in previous studies (*J. Am. Chem. Soc.* **2014**, 136, 13138–13141; *ACS Nano* **2020**, 14, 7847– 7859.), the RBCs were first fixed with 4% formaldehyde for 24h and then silicified. In this text, the blood samples were direct cryosilicified without any fixation. The ice recrystallization during the thawing process would damage the RBC membrane, causing the hemolysis (*J. Am. Chem. Soc.* **2019**, 141, 19, 7789–7796.). Notably, the damage of erythrocytes is irrelevant for DNA banking because they have no cell nucleus to store genetic information.

To cryosilicify cells, fresh anticoagulant peripheral blood was mixed with a pH 3 isotonic saline solution containing 15% w/w cryoprotectant hydroxyethyl starch and 50 mM silicic acid, i.e. Si(OH)₄ derived from the hydrolysis of tetramethyl orthosilicate (TMOS), at room temperature. The pH value of 3 was deliberately chosen, because then TMOS hydrolyses rapidly and forms mainly uncharged Si(OH)₄ monomers, while their condensation to siloxane oligomers is suppressed. After immersing cells into this silicification solution, the silicic acid species diffuse into the cell and successively exchange with water within hydrogen-bonded interfacial water networks surrounding biomolecular interfaces. The mixture was kept at -80 °C for 24 h, thawed at room temperature and rinsed with a phosphate buffered saline solution (Fig. 2a). During the freezing step, the formation of nanosized ice crystals permeabilizes the cellular membranes and Si(OH)₄ further concentrates, accelerating its adsorption at extra- and intracellular biomolecular surfaces. Thereafter, the amino acids of proximal proteins catalyse the progressive, self-limiting condensation of silicic acid. This ‘freezes’ the entire cell in nanoscopic amorphous silica coatings (Fig. 2b-e).

2. They mentioned the costly cryopreservation of extracted DNAs, but a control experiment with the extracted DNAs in silica matrices or silica-coated DNAs should be done. DNA extraction and subsequent silicification would not be highly costly.

Response: Many thanks for this suggestion. Following the suggestion, the preservation of whole blood DNA through encapsulating whole blood DNA in silica has been evaluated. The samples were prepared following previous study with slight modification (*Nat. Protoc.* **2013**, *8*, 2440–2448.). Instead of encapsulating small DNA fragments (100bp) as previous study, to match our objective of whole blood DNA preservation, whole blood DNA (8kbp) was encapsulated in silica. However, as shown in **Figure R1**, the genomic DNA from DNA encapsulation in silica vanished (blurred) even before storage. This unsuccessful DNA preservation might be due to our unskilled practice or the large DNA fragments. It worth to note that, our cryosilicification protocol itself only takes about 1 day for completion with a total hands-on time of less than 20 min. Additionally, our technology is not applied to purified DNA, but to the DNA still present in blood cells, which brings the advantage that the DNA does not have to be purified from the blood at the timepoint of collection, but rather at the timepoint of analysis. This has a cost advantage, if significantly more samples are collected than analyzed. Moreover, the chemical costs of our procedure for one person were about 0.2 \$ and a total cost was less than 0.5 \$. We believe this is a very competitive price for DNA banking.

3. It is not clearly discussed why and how the silica protects DNAs from degradation. In addition, the control experiment at room temperature should be done, like the previous report on proteins.

Response: Many thanks for this suggestion. DNA is susceptible to hydrolysis, depurination, depyrimidination, oxidation and alkylation and easily degrades resulting from the environmental stresses, including humidity, nucleases, mutagenic chemicals, ROS, and ionizing radiation (*Nat. Protoc.* **2013**, *8*, 2440–2448; *Angew. Chem. Int. Ed.* **2015**, *54*, 2552-2555; *Small* **2016**, *12*, 452-456; *Adv. Funct. Mater.* **2019**, *29*, 1901672.). In this work, mimicking the DNA in ancient fossils and bacterial spores, we created the siliceous cytoskeletons on whole blood samples after cryosilicification. The siliceous cytoskeletons could physically and chemically isolate DNA from these stresses, improving its long-term stability and integrity.

Following the suggestion of the reviewer, the control experiment at room temperature was carried out. As shown in **Figure S20**, there is no significant degradation of DNA from cryosilicified blood samples during the storage at room temperature for 68 days, supporting the long-term whole blood DNA preservation by cryosilicification. This information has been shown in the revised manuscript. (Please see page 12, paragraph 2, line 19-21)

4. HF is used for desilicification. DNA stability under the desilicification process needs to be discussed along with justification of the HF use. Also, it should be stated how DNAs are extracted from paper-based and 3D printing-based samples: the manuscript just says “was simply dissolved”, and the supplementary materials do not contain the information.

Response: Many thanks for this good question. The DNA is encapsulated in silicified cells. Without desilicification by buffered HF solution, it is impossible to extract DNA from silicified cells. In this text, all the DNA samples from silicified cells were prepared through desilicification and then extraction. In other words, the influence of desilicification has been already evaluated in all the analysis. As shown in **Figure S10**, after desilicification, 6.26 μ g DNA were extracted for cryosilicified blood samples, corresponding to 76.7% of the yield of fresh blood samples, which might be partially due to the damage caused by buffered HF solution. However, both gel electrophoresis of G6PD DNA fragments studies and whole genomic sequencing showed that the DNA of cryosilicified blood samples even through harsh desilicification process could still maintain its integrity comparable to those of fresh blood samples (**Figure 3b and Figure 3c**). These results demonstrated that buffered HF solution may not have a great impact on the DNA quality and stability. Similar phenomena were also observed in previous publications, where the DNA can be recovered from the glass spheres encapsulated DNA without harm by using buffered HF solutions (*Nat. Protoc.* **2013**, 8, 2440–2448; *Angew. Chem. Int. Ed.* **2015**, 54, 2552-2555; *Adv. Funct. Mater.* **2019**, 29, 1901672). To clarify our DNA extraction process, the details of DNA extracted from paper-based and 3D printing-based samples was added in the revised supplementary materials (Please see **section S25**).

We thank again the reviewers for expressing their comments and concerns and providing advice to improve the manuscript.

Sincerely,

Wei Zhu and Stefan Wuttke

REVIEWERS' COMMENTS

Reviewer #1 (Remarks to the Author):

The authors have performed a very substantial revision of the manuscript, and have included new data, which supports the conclusions of the work.

Prior to publication, however, the first (line 23-25) and last (line 371ff) sentence of the manuscript should be overworked, as neither is a scientific statement, but rather marketing and/or futurism.

Reviewer #2 (Remarks to the Author):

The authors have addressed my concerns in their revision and rebuttal.

Reviewer #3 (Remarks to the Author):

The revised manuscript (with the responses) has been read carefully, and it could be concluded that the work has some impact in the aspect of potential applications (not scientific originality).

Responses to reviewer's comments are as follows:

Reviewer #1:

The authors have performed a very substantial revision of the manuscript, and have included new data, which supports the conclusions of the work.

Prior to publication, however, the first (line 23-25) and last (line 371ff) sentence of the manuscript should be overworked, as neither is a scientific statement, but rather marketing and/or futurism.

Response: We thank this reviewer for their careful consideration of our MS and for providing very useful criticisms, suggestions, and recommendations. Following the suggestion of the reviewer, the related description has been changed in the revised manuscript. (Please see page 2, paragraph 1, line 1-2; page 19, paragraph 1, line 10-13)

Reviewer #2 (Remarks to the Author):

The authors have addressed my concerns in their revision and rebuttal.

Response: We thank again this reviewer for their careful consideration of our MS and for providing very useful criticisms, suggestions, and recommendations.

Reviewer #3 (Remarks to the Author):

The revised manuscript (with the responses) has been read carefully, and it could be concluded that the work has some impact in the aspect of potential applications (not scientific originality).

Response: We thank again this reviewer for their careful consideration of our MS and for providing very useful criticisms, suggestions, and recommendations.

We thank again the reviewers and editor for expressing their comments and concerns and providing advice to improve the manuscript.

Sincerely,

Wei Zhu and Stefan Wuttke